# Excessive excision of correct nucleotides during DNA synthesis explained by replication hurdles

Anupam Singh[1], Manjula Pandey[1], Divya Nandakumar[1], Kevin D Raney[2], Y Whitney Yin[3] & Smita S Patel[1,*]

## Abstract

The proofreading exonuclease activity of replicative DNA polymerase excises misincorporated nucleotides during DNA synthesis, but these events are rare. Therefore, we were surprised to find that T7 replisome excised nearly 7% of correctly incorporated nucleotides during leading and lagging strand syntheses. Similar observations with two other DNA polymerases establish its generality. We show that excessive excision of correctly incorporated nucleotides is not due to events such as processive degradation of nascent DNA or spontaneous partitioning of primer-end to the exonuclease site as a "cost of proofreading". Instead, we show that replication hurdles, including secondary structures in template, slowed helicase, or uncoupled helicase–polymerase, increase DNA reannealing and polymerase backtracking, and generate frayed primer-ends that are shuttled to the exonuclease site and excised efficiently. Our studies indicate that active-site shuttling occurs at a high frequency, and we propose that it serves as a proofreading mechanism to protect primer-ends from mutagenic extensions.

**Keywords** DNA polymerase; exonuclease activity; primer shuttling; replication hurdles; translocation

**Subject Category** DNA Replication, Recombination & Repair

**The EMBO Journal (2020) 39: e103367**

## Introduction

Replicative DNA polymerases (DNAPs) are responsible for faithfully copying the genomic DNA. These polymerases have a high selectivity for the correct incoming nucleotide, which keeps replication error rates low between $10^{-4}$ and $10^{-6}$ (Kunkel & Bebenek, 2000; Alba, 2001; Kunkel, 2004; Johansson & Dixon, 2013). Additionally, errors are efficiently removed by the proofreading 3′–5′ exonuclease activity, which increases accuracy by a factor of 10- to 1,000-fold (Capson et al, 1992; Reddy et al, 1992; Johnson, 1993; Shcherbakova et al, 2003; Kunkel, 2004; Reha-Krantz, 2010). Thus,

mutations in the exonuclease site (Exo-site) result in replication errors, including base substitution, insertions, and deletions (Muzyczka et al, 1972; Cotterill et al, 1987; Reha-Krantz, 1998; Shcherbakova et al, 2003). Exo-site mutants of replicative Pol ε and Pol δ DNAPs are found in tumors, and many of them show an ultramutational phenotype (Albertson et al, 2009). Interestingly, the ultramutational phenotype is not explained entirely by the loss of proofreading activity (Kane & Shcherbakova, 2014; Li et al, 2018), suggesting that there are additional roles of the exonuclease activity that prevent mutagenicity.

The polymerase and exonuclease active sites in replicative polymerases are typically located 30–40 Å apart (Beese et al, 1993; Doublie et al, 1998); however, the primer-end can melt and reach the Exo-site and thus shuttles between the two active sites (Beechem et al, 1998; Berezhna et al, 2012). Once the primer-end is in the Exo-site, the 3′-end nucleotide is excised irrespective of its origin from a correct or an incorrect nucleotide addition reaction. Selective removal of the wrong versus right nucleotide is explained by the kinetic partitioning model (Donlin et al, 1991; Johnson, 1993). According to this model, the primer-end partitions to the Exo-site primarily during a misincorporation event. The observed rate of excision of a matched primer-end is slower than a mismatched primer-end due to its more delayed transfer to the Exo-site. Based on the excision rates of correctly base-paired 3′-end nucleotide, this partitioning model predicts that only 0.1% of the correctly incorporated nucleotides will be excised during DNA synthesis.

However, a study by Fersht et al suggested that DNA polymerase I from Escherichia coli excises 7–15% of correct nucleotides during ongoing DNA synthesis (Fersht et al, 1982). Polymerase fidelity is exceptionally high, and thus, misincorporation events are infrequent during synthesis through normal DNA; therefore, it was puzzling why a significant amount of nucleotides was excised during DNA synthesis. Fersht explained the excessive excision of correctly incorporated nucleotides as the "cost of proofreading" that an accurate polymerase must pay to maintain a high level of fidelity. However, the mechanistic basis for the excessive excision remained unknown and controversial since the partitioning model predicted only 0.1% excision of correct nucleotides during normal DNA synthesis. Recent

1 Department of Biochemistry and Molecular Biology, Robert Wood Johnson Medical School, Rutgers University, Piscataway, NJ, USA
2 Department of Biochemistry and Molecular Biology, The University of Arkansas for Medical Sciences, Little Rock, AR, USA
3 Department of Pharmacology and Toxicology, Sealy Center for Structural Biology, University of Texas Medical Branch, Galveston, TX, USA
*Corresponding author. Tel: +1 732 235 3372; E-mail: patelss@rutgers.edu

single-molecule studies also showed that a significant percentage of correct nucleotides are excised and suggested that this occurs in conjunction with mismatch removal because the Exo-activity continues to degrade the nascent DNA processively after removing the mismatch (Hoekstra *et al*, 2017).

T7 replisome represents an attractive model system to understand the underlying mechanisms for accurate DNA replication, because of its simplicity in terms of the number of proteins required to constitute the replisome and its high efficiency and fidelity of DNA synthesis (Hamdan & Richardson, 2009). During our studies of the T7 DNAP, we also happened to observe excessive excision of correct nucleotides, which raised our curiosity to understand the mechanistic basis of this phenomenon. Our measurements indicated that T7 replisome hydrolyzes close to 7% of correctly incorporated nucleotides during leading and lagging strand DNA synthesis, which clearly cannot be explained by misincorporation events. Two other replicative DNAPs showed similar amounts of nucleotides excised during rolling circle DNA synthesis, which demonstrates that this phenomenon is not peculiar to T7 replisome. After an extensive mechanistic study, we concluded that excessive excision of correctly incorporated nucleotide is not due to processive excision of nascent DNA or due to spontaneous partitioning of the correctly base-paired primer-end to the Exo-site as the cost of proof-reading, as suggested previously. Instead, excessive excision results from shuttling of the primer-end to the Exo-site triggered by translocation problems such as secondary structures in DNA and helicase–polymerase uncoupling during DNA synthesis. We find it remarkable that these events are not rare and contribute in a significant way to excision of correctly incorporated nucleotides during normal DNA synthesis.

## Results

### Excessive excision of correct nucleotides during leading and lagging strand replicative DNA synthesis

T7 replisome is one of the simplest replicative complexes that can be assembled *in vitro* (Hamdan & Richardson, 2009). Only four proteins are needed to reconstitute leading and lagging strand DNA synthesis: T7 gp4 has helicase and primase activities (referred to as T7 helicase), T7 gp5 has polymerase and 3′–5′ exonuclease activities and uses *E. coli* thioredoxin as a processivity factor (gp5:thioredoxin is referred to as T7 DNAP), and T7 gp2.5 is the single-stranded binding protein. To simultaneously measure the polymerase (Pol) and exonuclease (Exo) activities, we chose a 70 bp minicircle DNA annealed to a 110-mer primer that supports efficient rolling circle leading and lagging strand DNA synthesis (Lee *et al*, 1998; Pandey *et al*, 2009). Previous studies that measured Pol- and Exo-activities used linear primer–templates where the polymerase would idle at the 3′-end of the primer when it runs out of template and produce the dNMP excision products (Mizrahi *et al*, 1986; Pavlov *et al*, 2004). The use of circular minicircle template in the present study eliminates the end-idling problem and provides an accurate measurement of the Exo-activity during active DNA synthesis. Additionally, the 70 bp minicircle DNA contains a priming sequence for the T7 primase that makes short RNA primers in the presence of ATP and CTP, which are elongated by T7 DNAP to

make the Okazaki lagging strand DNA products. Due to skewed G and C contents in leading and lagging strand DNA products, we can measure leading strand synthesis nearly exclusively by using $\alpha$-$^{32}$P-dGTP and lagging strand synthesis by using $\alpha$-$^{32}$P-dCTP.

To measure the Pol- and Exo-activities during leading strand synthesis, we spiked the nucleotide mix with $\alpha$-$^{32}$P-dGTP. Incorporation of $\alpha$-$^{32}$P-dGMP in the growing DNA provides an estimate of the Pol-activity, and production of free $\alpha$-$^{32}$P-dGMP nucleotide provides the Exo-activity (Fig 1A and B). Since Pol- and Exo-activities are measured in the same reaction, the ratio of the two activities (Pol/Exo) provides the average number of excision events occurring relative to the polymerization events.

We measured Pol- and Exo-activities at a constant 500 µM concentration of dTTP and 50–300 µM concentration of dVTPs (a mixture of dATP, dGTP, and dCTP) (Fig 1C and D, Appendix Fig S1A). The dTTP concentration was maintained at a higher level because T7 helicase prefers to use dTTP as fuel for DNA unwinding to support efficient leading strand synthesis. The Pol- and Exo-activity kinetics during rolling circle leading strand synthesis were linear indicating that we were measuring ongoing DNA synthesis and excision activities (Fig 1C and D). Surprisingly, the ratio of the two activities, Pol/Exo ratio, was 15 ± 2, which indicates that T7 DNAP, on an average, excises one nucleotide for every 15 nucleotides incorporated into the DNA (Fig 1E). We observed a similar Pol/Exo ratio of 12 ± 0.2 with the $\alpha$-$^{32}$P-dATP tracer (Appendix Fig S1B), which indicates that the type of radioactive nucleotide does not influence the relative activities. Moreover, the Pol/Exo ratio of about 15 remained constant over the range of dVTPs concentration (Fig 1E). A detailed analysis described below provided the significance of this result.

To verify that the excision was occurring during processive DNA synthesis and was not due to protein dissociation and rebinding events, we added a protein trap (35/68 mer primer–template with 3′dideoxy primer-end) 30 s after starting the rolling circle reaction with the T7 replisome. The trap with the 3′ddNMP end is an effective trap that would stably remove free and newly dissociated DNAPs without itself being a source of excision products. The Pol/Exo ratio remained unchanged after addition of the trap (Figs 1F and EV1A). The unchanged Pol/Exo ratio indicates that excessive excision is occurring during processive DNA synthesis; hence, the primer-end transfer from the Pol-site to the Exo-site is intramolecular.

To determine whether coupling of the leading and lagging strand syntheses affects the Pol/Exo ratio, we carried out reactions in the presence of ATP and CTP nucleotides that prime Okazaki DNA synthesis. The Pol/Exo ratios of coupled and uncoupled leading strand synthesis reactions were similar (Fig 1F). We also measured the Pol/Exo ratio of lagging strand synthesis reaction alone by spiking the dNTPs with $\alpha$-$^{32}$P-dCTP (Figs 1G and EV1Bi). Pol-activity was measured from the incorporation of $\alpha$-$^{32}$P-dCMP in the DNA and Exo-activity from the production of $\alpha$-$^{32}$P-dCMP (Fig EV1Bii). The Pol/Exo ratio during lagging strand synthesis was 16 ± 3 (Fig 1H), which is similar to the ratio observed during leading strand synthesis. We believe that the similarity in the ratios is coincidental and they could be different with a minicircle DNA of a different sequence.

Since these results were unexpected, we wanted to be sure that the exonuclease activity of T7 DNAP was responsible for the observed dGMP in our reactions; hence, we carried out several

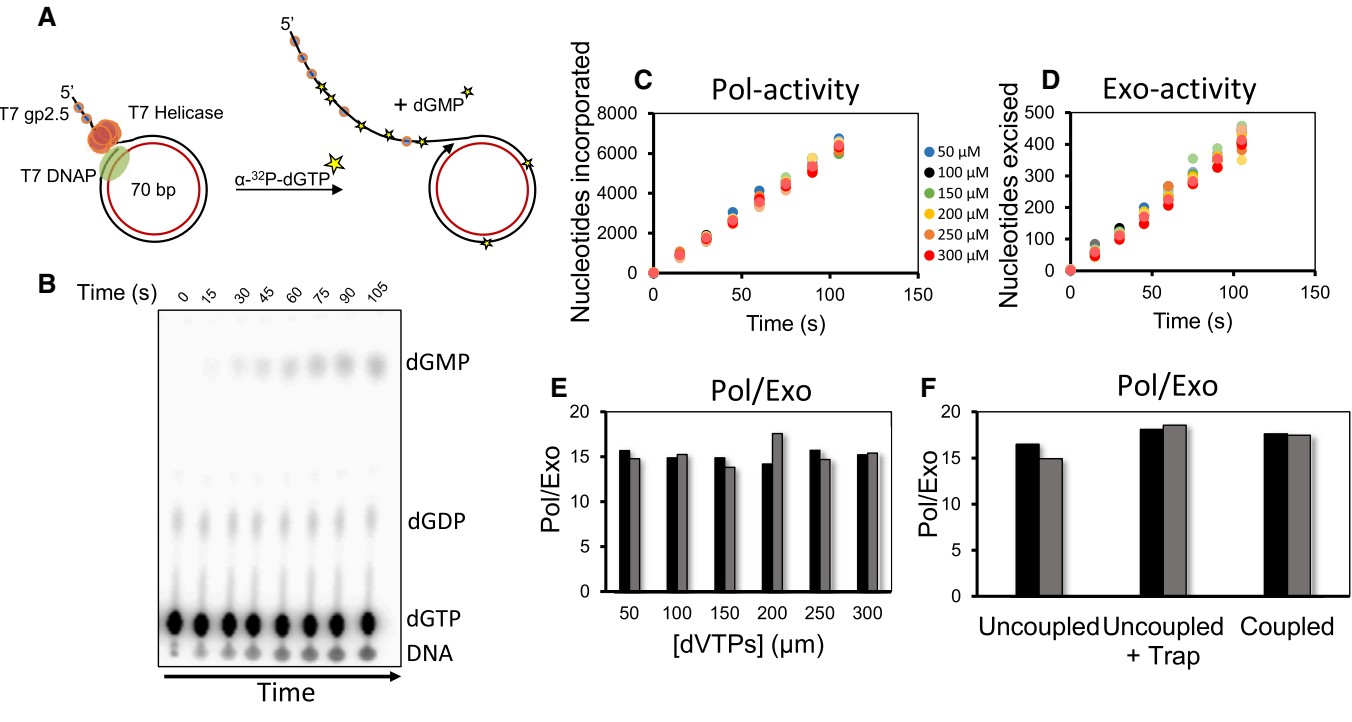

## Lagging strand DNA synthesis

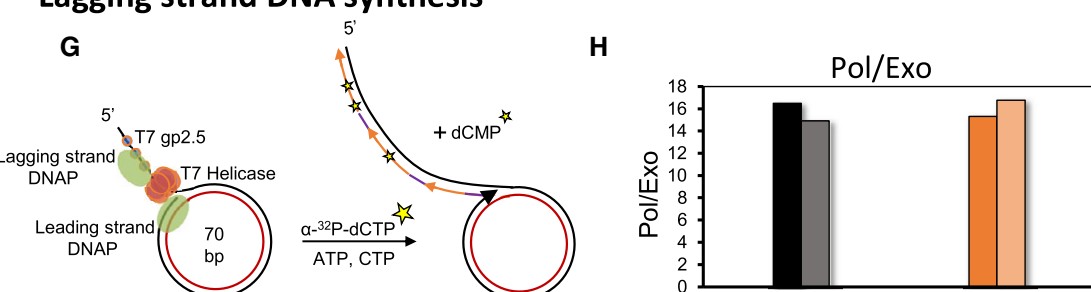

**Figure 1. Polymerase and exonuclease activities of T7 DNAP measured during ongoing DNA synthesis.**

A    The experimental design to simultaneously measure Pol- and Exo-activities during leading strand synthesis. T7 replisome was assembled on a minicircle DNA substrate that supports efficient rolling circle leading strand synthesis. Asterisks represent incorporated and excised radiolabeled nucleotides.

B    Representative TLC image shows the locations of nascent DNA from the Pol-activity, remaining dGTP substrate, dGDP contamination, and excised dGMP from the Exo-activity.

C, D    Time courses of Pol-activity (C) and Exo-activity (D) expressed as moles of nucleotides incorporated and excised per mole of substrate DNA used. Experiments were performed at a constant 500 μM dTTP concentration and increasing concentration of dVTPs.

E    Pol/Exo ratios were calculated from the slopes of linear fits of data in (C and D).

F    Pol/Exo ratio during leading strand synthesis in the absence and presence of a protein trap and leading strand synthesis coupled with lagging strand synthesis. 800 nM trap was added at 30 s after reaction initiation. Coupled synthesis reactions were carried out in the presence of priming nucleotides, ATP and CTP.

G    Lagging strand reactions were performed in the presence of ATP and CTP and α-$^{32}$P-dCTP.

H    Pol/Exo ratio during leading and lagging strand reactions is compared. Pol/Exo-data for uncoupled leading strand synthesis from (F) are used here for comparison.

Data information: Circles in two shades of the same color show data from two individual experiments. Bars represent data from two individual experiments.

control experiments. We used the exonuclease-deficient mutant of gp5 (exo− T7 gp5 D5A, E7A, purified using the same protocol as exo+ T7 gp5) and did not detect dGMP (Fig EV2A), which rules out contaminating exonuclease activity in the protein samples. Furthermore, we observed 3% excision of newly incorporated nucleotides during DNA synthesis on a primed M13 ssDNA template (see

below), which ruled out minicircle as a peculiar template. There was no dGMP produced when DNA synthesis was inhibited by leaving out the DNA primer or nucleotides (Fig EV2B), which ruled out contaminating dGTPase in our reactions. We conclude that the source of α-$^{32}$P-dGMP in our reactions is from the 3′–5′ exonuclease activity of T7 DNAP.

## Excessive nucleotide excision is observed with other replicative DNAPs

To test the generality of this excessive excision activity, we carried out similar reactions with two other replicative DNA polymerases, *Saccharomyces cerevisiae* mitochondrial DNAP, Mip1, and bacteriophage Φ29 DNAP which belong to A-family and B-family of polymerases, respectively. Rolling circle DNA synthesis reaction was carried out using the minicircle DNA in the presence of all dNTPs and $\alpha$-$^{32}$P-dGTP (Fig 2A). T7 DNAP requires T7 helicase and T7 gp2.5 to catalyze strand displacement DNA synthesis. Φ29 DNAP exhibits robust strand displacement DNA synthesis on its own, and Mip1 can catalyze strand displacement synthesis in the presence of Rim1, the yeast mitochondrial single-stranded DNA binding protein (Rodriguez *et al*, 2005; Viikov *et al*, 2011). Efficient incorporation of dGMP in the DNA was observed with all three DNAPs, and in all cases, there was a high amount of excised dGMP in the reaction products (Fig 2B), which yielded average Pol/Exo ratio of $15 \pm 0.2$ for T7 replisome, $9 \pm 1.3$ for Mip1, and $41 \pm 1$ for Φ29 DNAP (Fig 2C). These experiments demonstrate that excessive excision of incorporated nucleotides is a general feature of replicative DNA polymerases.

The Pol/Exo ratio of ~15 corresponds to excision of about 7% of incorporated nucleotides, which is remarkable. Based on the known error frequency of the exonuclease-deficient mutants of DNAPs ($10^{-5}$–$10^{-6}$), we expect undetectable amounts of excised nucleotides from misincorporation events (< 0.001%). Therefore, the high number of excision events cannot be from proofreading of misincorporated nucleotides and likely represents correct nucleotide excision.

## Excessive excision is due to the frequent shuttling of primer-end between Pol- and Exo-sites

To understand the mechanistic basis for excessive excision of correctly incorporated nucleotides during DNA synthesis, we considered several possibilities. It is known that primer-end can travel between the Pol-site and the Exo-site (Berezhna *et al*, 2012), but it is thought that this happens mainly after misincorporation. Several single-molecule magnetic and optical tweezer experiments have observed that when the primer-end is transferred to the Exo-site under conditions mimicking a misincorporation, there is excision of hundreds of nucleotides of nascent DNA before the primer-end returns to the Pol-site (Wuite *et al*, 2000; Ibarra *et al*, 2009; Hoekstra *et al*, 2017). Thus, the source of excised nucleotides could be from processive degradation of nascent DNA after the mismatch is removed. Alternatively, it is possible that primer-end is shuttled more often than expected during ongoing DNA synthesis and these events are responsible for removing correctly incorporated nucleotides.

To distinguish between the above two possibilities (Fig 3A), we used the following experimental strategy. It is known that the $R_p$ phosphorothioate linkage (Fig 3B) is resistant to the 3′–5′ exonuclease activity (Kunkel *et al*, 1981; Brody *et al*, 1982; Eckstein, 1985). We verified this using a commercially synthesized DNA primer with a phosphorothioate linkage next to a mismatched primer-end. The primer–template with the phosphorothioate linkage was more resistant to excision by exo+ T7 DNAP than the normal DNA (Fig EV3A).

Therefore, we could use phosphorothioate linkage as a molecular brake to distinguish between the two types of exonuclease activity.

To create nascent DNA with the phosphorothioate linkages, we added $\alpha$-thio-dNTP (dNTP$\alpha$S) in our reactions (Fig 3C). Commercially available dNTP$\alpha$S is a racemic mixture of $R_p$ and $S_p$ dNTP$\alpha$S, but DNAPs incorporate only the $S_p$ dNTP$\alpha$S (Eckstein, 1985). Incorporation of $S_p$ dNTP$\alpha$S is accompanied by a stereochemical inversion to generate an $R_p$ phosphorothioate linkage (Brody *et al*, 1982), which is resistant to the exonuclease activity. The mismatched primer–template with the phosphorothioate linkage was more resistant to excision by exo+ T7 DNAP than the normal DNA (Fig EV3A). If hundreds of correct nucleotides in the nascent DNA are removed processively when the primer-end is in the Exo-site, then the phosphorothioate linkages in the nascent DNA will act as a molecular brake to inhibit the Exo-activity, and Pol/Exo ratio will be much higher in the presence of dNTP$\alpha$S in comparison with normal dNTPs. On the other hand, if only one or two nucleotides from the primer-end are removed when the primer-end is in the Exo-site, then the presence of dNTP$\alpha$S will have a negligible effect on the Pol/Exo ratio.

Rolling circle DNA synthesis reactions were carried out with T7 replisome wherein one dNTP at a time was replaced with dNTP$\alpha$S (except dGTP$\alpha$S). T7 helicase uses both dTTP and dTTP$\alpha$S equally as fuel for DNA unwinding (Fig EV3Bi and Bii). This allowed us to perform experiments with dTTP$\alpha$S. The Pol- and Exo-activities were measured by spiking the nucleotide mix with $\alpha$-$^{32}$P-dGTP. The Pol-activity was 25–50% lower in the presence of dNTP$\alpha$S than in the presence of normal dNTPs (Figs 3D and EV3Biii), which was reported previously (Pugliese *et al*, 2015). The Exo-activity, however, was reduced equally (Figs 3E and EV3Biv); hence, the resulting Pol/Exo ratios in each of the thio-dNTP reactions were similar to the control reaction with standard nucleotides (Fig 3F). These results demonstrate that excessive excision activity does not arise from processive degradation of hundreds of nucleotides in the nascent DNA. We conclude that excessive excision of correctly incorporated nucleotides is due to frequent shuttling of the primer-end from the Pol-site into the Exo-site during DNA synthesis.

## Primer-end binding to the Exo-site initiates from the pre-translocation state

What events actuate frequent shuttling of the primer-end into the Exo-site during ongoing DNA synthesis? To gain further insights into the mechanism of active-site switching, we asked at what step in the nucleotide incorporation cycle the primer-end is shuttled to the Exo-site. It has been suggested that primer-end transfer to the Exo-site initiates from the pre-translocation state (Lieberman *et al*, 2014). We, therefore, designed the following experiments to test this model.

Inorganic pyrophosphate (PPi) is a byproduct of the nucleotide incorporation reaction during DNA synthesis. PPi binds to the pre-translocation state of the DNAP and catalyzes the pyrophosphorolysis reaction to shorten the primer-end by one nucleotide (Fig 4A). Thus, pyrophosphorolysis is commonly used to assess the translocation state of RNA polymerases and reverse transcriptases (Marchand & Gotte, 2003; Hein *et al*, 2011). It is known that the DNA sequence itself can influence the equilibrium between pre-translocation state and post-translocation states (Tabor & Richardson, 1990;

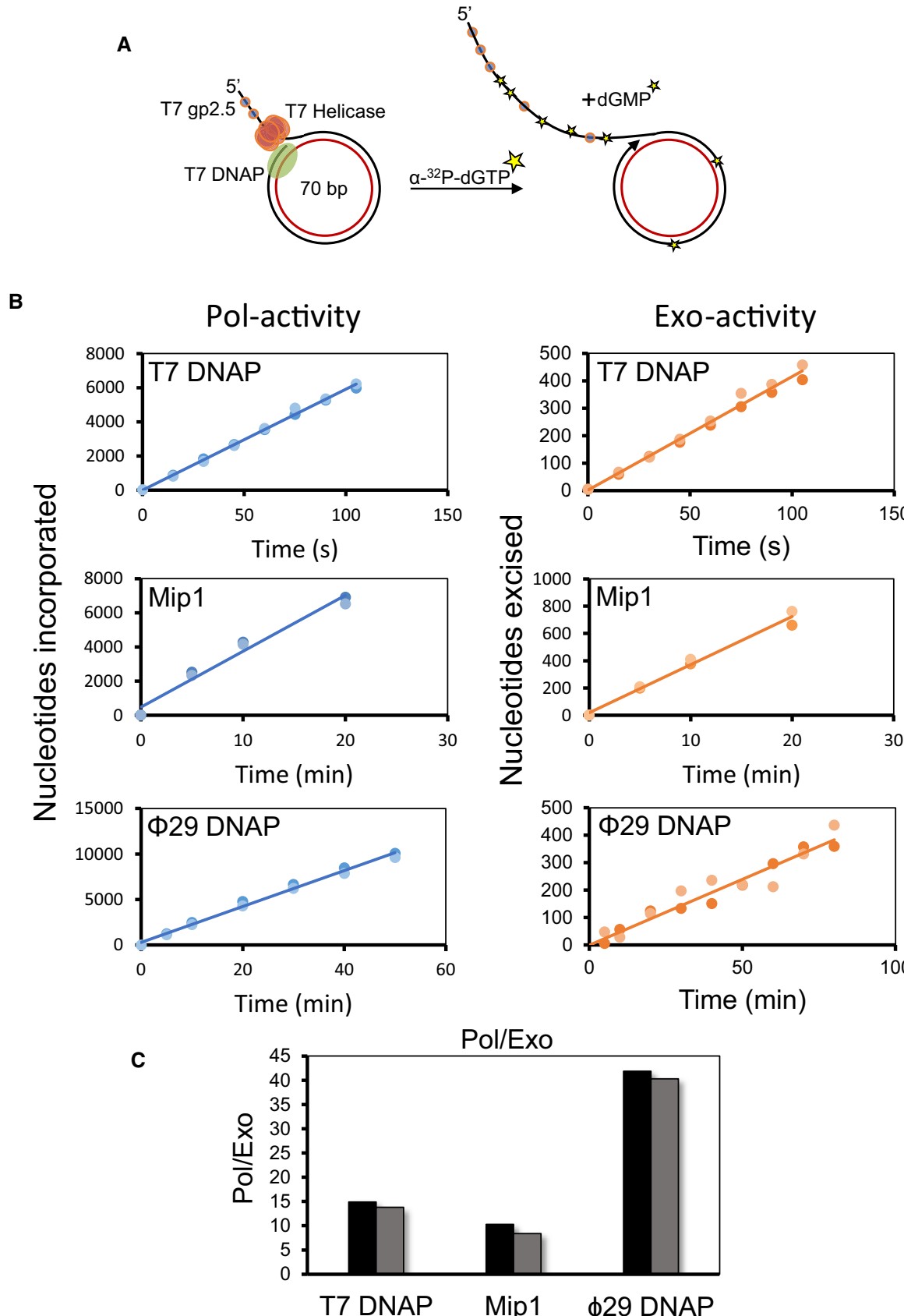

**Figure 2.**

**Figure 2.** **Excessive excision of incorporated nucleotides during rolling circle DNA synthesis is a general feature of replicative DNAPs.**

A Experimental design to simultaneously measure Pol-activity and Exo-activity during rolling circle DNA synthesis. Reactions with T7 DNAP were performed as described in Fig 1A. Reactions with Mip1 were carried out in the presence of mitochondrial SSB (Rim1), and Φ29 DNAP reactions did not contain any SSB protein. Reactions were performed with 150 μM dNTPs (150 μM dVTPs and 500 μM dTTP in case of T7 DNAP).

B Nucleotides incorporated (left panels) and excised (right panels) during rolling circle DNA synthesis by T7 DNAP, Mip1, and Φ29 DNAP. Y-axes represent moles of nucleotides incorporated or excised per mole of the DNA substrate used.

C Pol/Exo ratio determined from the slopes of linear fits of data presented in (B). Pol/Exo-data for T7 DNAP from Fig 1E (150 μM dVTPs) are reused here for comparison.

Data information: Circles represent data from two individual experiments. Bars show data derived from two individual experiments.

## Processive versus distributive excision

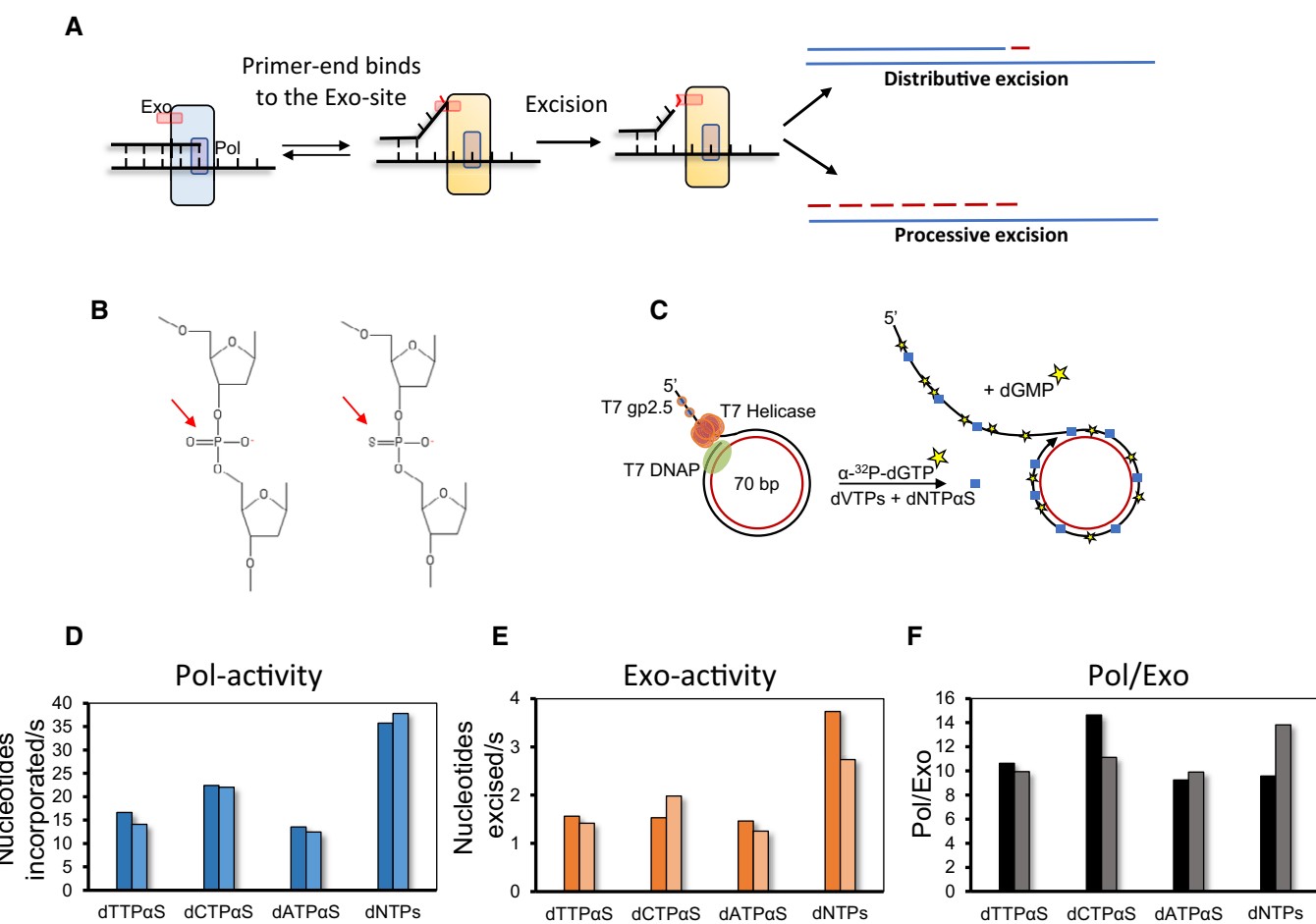

**Figure 3.** **Excessive excision of incorporated nucleotides is not due to processive degradation of the nascent DNA.**

A Two possible ways to generate dNMPs during ongoing DNA synthesis. In a processive excision mode, the primer-end transferred from the Pol-site remains at the Exo-site for a prolonged duration, which results in excision of hundreds of nucleotides. In the distributive excision mode, dNMPs are generated from the frequent shuttling of the primer-end between Pol-site and Exo-site and excision of one nucleotide per shuttling event.

B Chemical structures of phosphodiester (left) and phosphorothioate (right) linkages in the DNA strand.

C The experimental design to discriminate between processive excision and distributive excision modes of dNMP generation. The blue square represents thio-nucleotide incorporated in the DNA.

D, E Pol- and Exo-activities are shown as moles of nucleotides incorporated and excised per mole of substrate DNA per second.

F Pol/Exo ratio determined from Pol- and Exo-activities in (D and E).

Data information: Bars show data from two individual experiments.

Reha-Krantz *et al*, 2014), and thus, the pyrophosphorolysis rate constant is sensitive to the particular base-pair at the primer-end (Donlin *et al*, 1991). If primer-end binds to the Exo-site primarily from the pre-translocation state of the DNAP, then there will be a direct correlation between the rates of the exonuclease and pyrophosphorolysis activities.

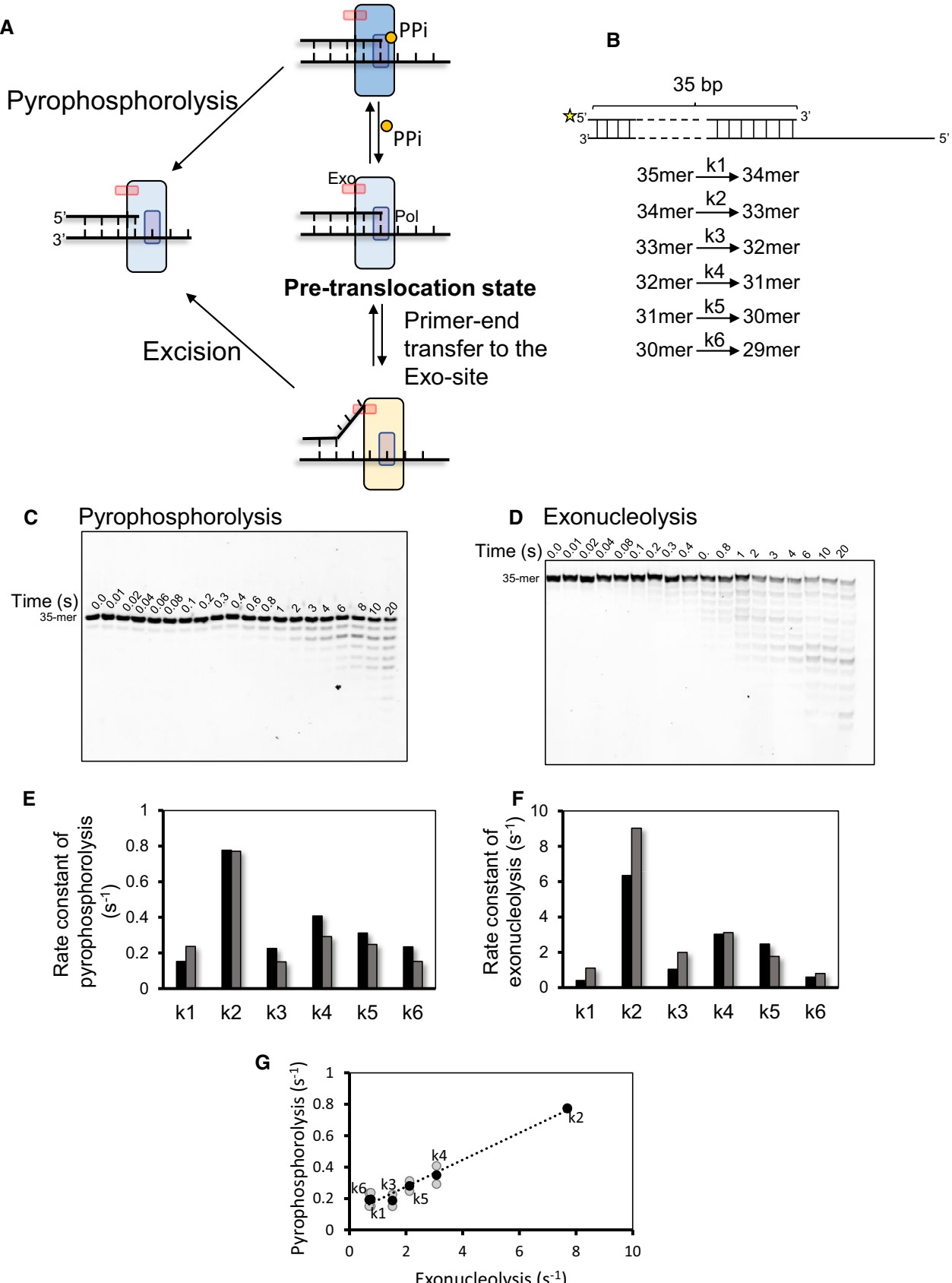

Figure 4.

◄

**Figure 4.  Primer-end transfer from polymerase to exonuclease site initiates from the pre-translocation state.**

A    Schematic of DNAP catalyzing pyrophosphorolysis and exonuclease reactions from the pre-translocation state. PPi (yellow circle) binding to the pre-translocation state in the "fingers-closed" conformation (dark blue DNAP) results in dNTP synthesis and shortening of the primer-end by one nucleotide. The primer-end can be transferred to exonuclease site in the "fingers-open" conformation (light blue DNAP), resulting in excision reaction and shortening of the primer-end by one nucleotide.

B    Primer–template DNA substrate used for determining the pre-steady-state kinetics of pyrophosphorolysis and exonucleolysis. The asterisk at 5′ end of the primer is fluorescein label.

C, D    Pyrophosphorolysis reactions performed with exo– T7 DNAP and pyrophosphate (C) and excision reactions performed with exo+ T7 DNAP (D) quenched at stated time points and analyzed on 24% denaturing polyacrylamide gels.

E, F    Rate constants for pyrophosphorolysis (E) and excision reactions (F).

G    Correlation between rates of pyrophosphorolysis and exonucleolysis.

Data information: Gray circles are data points from two individual experiments. Black circles show the mean data. Bars show data from two individual experiments.

Pyrophosphorolysis and excision reactions were performed with the exo- and exo+ T7 DNAPs, respectively, on a primer–template substrate under single-turnover conditions using a rapid quenched-flow instrument. Both reactions progressively shorten the 35-mer primer to 29-mer and beyond with increasing reaction times. After resolving the DNA products on a polyacrylamide sequencing gel (Fig 4C and D), we quantified the DNA bands and fitted the single-turnover kinetics of each shortening reaction from 35-mer to 29-mer to the model in Fig 4B, which provided the intrinsic pyrophosphorolysis and excision reaction rate constants (Fig EV4A). The pyrophosphorolysis reaction occurs with a different rate constant at each nucleotide position (Fig 4E). Interestingly, the exonuclease rate constant varied in the same way and showed the same trend (Fig 4F). A comparison of the rate constants showed a direct correlation between the two activities (Fig 4G). These data are consistent with the model that primer-end binding to the Exo-site occurs from the pre-translocation state.

Similar experiments on a primer–template with an abasic site showed a correlation between pyrophosphorolysis and excision activities. A primer–template substrate with an abasic templating nucleotide at n+1 (with primer-end nucleotide as n) had a drastically reduced pyrophosphorolysis rate compared to the unmodified primer–template (Fig EV4B), which indicated that the abasic site destabilizes the pre-translocated state of the DNAP. We found that the excision rate of the primer-end was reduced similarly relative to the normal DNA (Fig EV4B). Thus, a lesion that destabilized the pre-translocated state decreased pyrophosphorolysis and excision activities to similar extents, demonstrating in a different way that primer-end shuttles to the Exo-site from the pre-translocated state of the DNAP.

**Excessive excision of correctly incorporated nucleotides is due to translocation problems**

Based on the measured excision rates of correctly base-paired primer-end, spontaneous partitioning of the correctly base-paired primer-end into the Exo-site should be a rare event. However, during ongoing DNA synthesis, the DNAP visits the pre-translocated states at every nucleotide incorporation cycle. Hence, partitioning of the correctly base-paired primer-end to the Exo-site could be more frequent than that predicted from the excision rate of correctly base-paired primer-end in the absence of DNA synthesis. Alternatively, since partitioning occurs from the pre-translocation state, translocation problems could increase the occupancy of these states to elevate the frequency of primer-end partitioning into the Exo-site.

To distinguish between spontaneous partitioning versus translocation problem-mediated partitioning models, we measured the Exo-activity as a function of increasing dNTP concentrations. If excessive excision activity results from spontaneous partitioning of the primer-end to the Exo-site during normal DNA synthesis, then the Exo-activity should be sensitive to dNTP levels. This is because the dNTP binding event stabilizes the post-translocated state and hence decreases the occupancy of the primer-end in the pre-translocated state. Therefore, spontaneous partitioning model predicts a decrease in Exo-activity with increasing concentration of dNTPs.

To test the model, DNA synthesis reactions were carried out at increasing dNTP conditions using M13 ssDNA template annealed to a short DNA primer. Unlike the minicircle assays where we kept dTTP at a high level for efficient helicase activity, this setup does not require T7 helicase; hence, we could vary all dNTP concentrations equally. Moreover, synthesis on gp2.5 covered ssDNA mimics lagging strand synthesis. Reactions were carried out using 1–30 μM dNTPs spiked with $\alpha$-$^{32}$P-dGTP (Fig 5A). In this range of dNTP concentration, Pol-activity increased in a hyperbolic manner (Fig 5B and D) with dNTP concentration providing an observed dNTPs $K_m$ of 7.9 ± 1.6 μM. Surprisingly, the Exo-activity increased slightly from 1 to 2 μM dNTPs but mostly remained constant and showed no decrease up to 30 μM dNTPs (Fig 5C and E). These results are inconsistent with the spontaneous partitioning model.

We used the Kintek Explorer kinetic modeling software (Johnson, 2009; Johnson et al, 2009b) to model the Pol-activity and Exo-activity at increasing dNTP concentrations. To model processive primer-end extension reaction (Fig 7, blue box), we assumed step-wise nucleotide addition to the primer-end in three elementary steps consisting of dNTP binding ($K_d$ of 20 μM), the conformational change (600/s), and nucleotide incorporation (300/s) (Fig EV5). These rate constants have been determined by previous pre-steady state kinetic studies of T7 DNAP (Johnson, 2010). To model the spontaneous partitioning of the primer-end to the Exo-site, we assumed that after each nucleotide incorporation step when the primer-end ends up in the pre-translocated state, it has a choice of partitioning to the post-translocated state or partitioning to the Exo-site. We assumed that translocation is a rapid step with an equilibrium constant of one, as suggested by nanopore experiments (Lieberman et al, 2014). We also assumed that the primer-end partitioning was the rate-limiting step and consistent with the observed rate of correct nucleotide excision, but once the primer-end is in the Exo-site, the excision activity is fast (Donlin et al, 1991).

The above-described mechanism (Fig 7 in the blue box) accurately predicted the hyperbolic increase in Pol-activity with

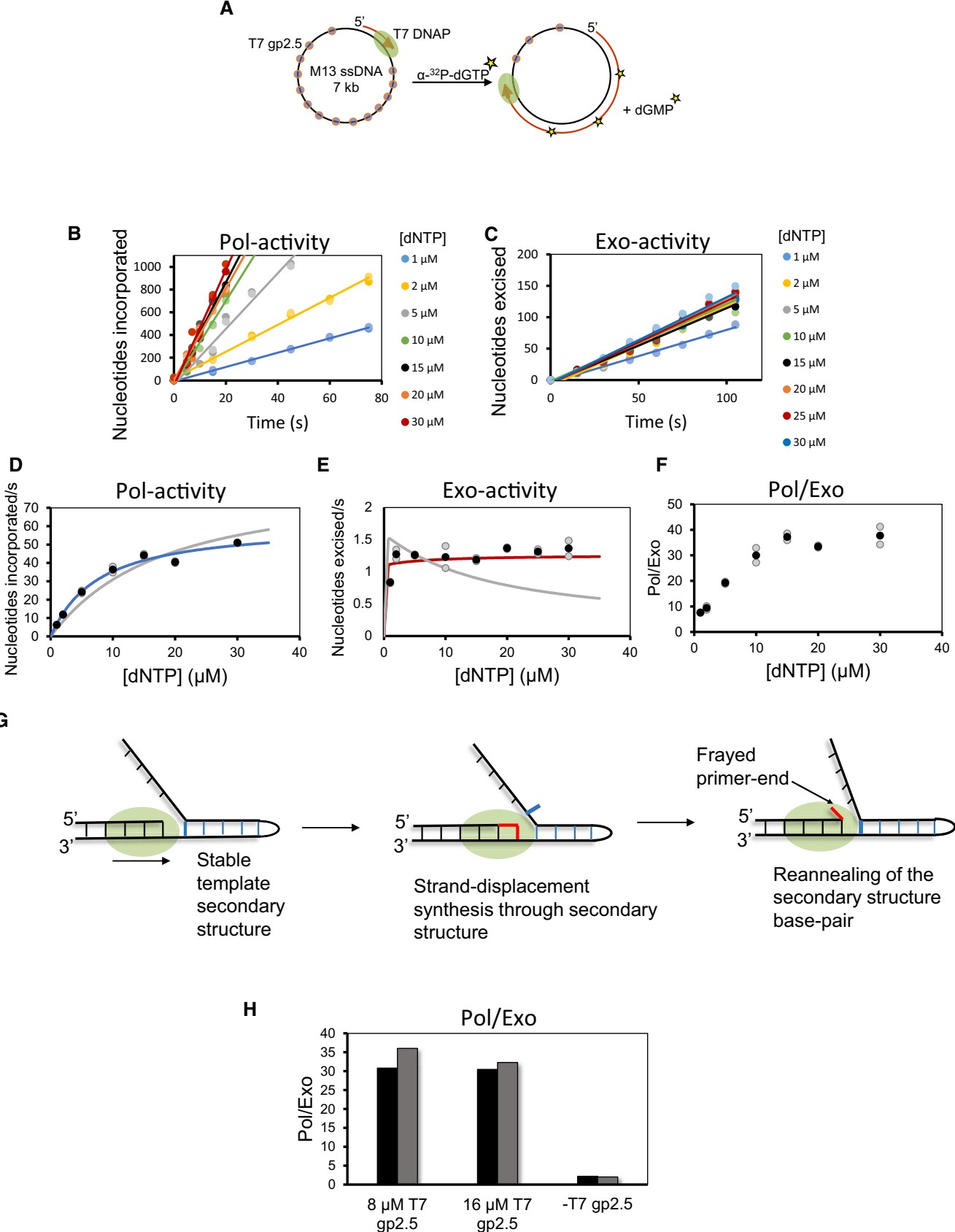

**Figure 5.**

◀

**Figure 5. Excessive excision of correctly incorporated nucleotides during DNA synthesis is due to translocation problems.**

A    Experimental design to simultaneously measure Pol- and Exo-activity of T7 DNAP with gp2.5 coated M13 ssDNA substrate annealed to a short DNA primer. Primer extension was carried out in the absence of T7 helicase, and reaction was traced with $\alpha$-$^{32}$P-dGTP (asterisks show radiolabeled nucleotides). Reactions were quenched and spotted on TLC to determine nucleotides incorporated and excised.

B, C    Moles of nucleotides incorporated (B) and excised (C) per mole of substrate DNA in reactions carried out with different dNTP concentrations. Circles in two shades of the same color show data points from two individual experiments. Lines show the linear fits of the mean data.

D, E    Nucleotides incorporated (D) and excised (E) per second (black circles) determined from the slopes of linear fits of data in (B, C). Blue- and red-colored curves are trends from the global fitting of Pol- and Exo-data to the complete model shown in Figs 7 and EV5. Gray curve represents the poor fit of data to the model in Figs 7 and EV5 within the blue box. Gray circles are data points from two individual experiments. Black circles show the mean data that were fitted to the models.

F    dNTP concentration dependence of Pol/Exo ratio (determined from D and E). Gray circles are data points from two individual experiments. Black circles show the mean data.

G    Model is showing template secondary structure as a translocation hurdle. A stable secondary structure can result in primer-end fraying.

H    Pol/Exo ratio during primer extension reactions performed with T7 DNAP in the presence and absence of T7 gp2.5. Reactions were carried out on primed M13 ssDNA substrate with 30 μM dNTPs. Bars show data from two individual experiments.

increasing dNTP concentration (Fig 5D). However, as expected, the mechanism predicted a decrease in the Exo-activity with increasing dNTP concentration, which is inconsistent with the experimental data (Fig 5E).

Next, we outlined an alternative model that invokes that primer-end partitions into the Exo-site when there are translocation problems. To model these occasional translocation problems, we created a branched pathway from the normal to a Pol-inhibited state (red arrow, Fig 7). The Pol-inhibited state is incompetent in DNA synthesis but competent in partitioning the primer-end to the Exo-site and excision activity. This Pol-inhibited model accurately predicted both Pol-activity and Exo-activity trends as a function of increasing dNTP concentrations (Fig 5D–F).

To determine which of the intermediates in the normal nucleotide incorporation cycle branch to the Pol-inhibited state, we used an iterative process to obtain best fits by trying different steps for branching. Best fits indicated that the Pol-inhibited state is populated both from the pre-translocation state and from the dNTP-bound post-translocated state. The range of the kinetic parameters for the partitioning steps was determined within the chi-square threshold limit of 0.67 set in FitSpace module of the Kintek Explorer software (Johnson *et al*, 2009a). Our data are best explained with rates of transfer from pre-translocation and dNTP-bound states in the range of 0.004 to 7/s and 3–12/s, respectively (Fig EV5). Further constraints from additional data sets can narrow the range for these parameters.

What type of translocation hurdles creates such Pol-inhibited states during DNA synthesis on the M13 ssDNA template? It is well known that template base lesions or mismatches can create translocation hurdles. However, the M13 ssDNA template lacks such lesions and misincorporation events are also rare. Hence, the presence of lesions or mismatches is not the reason for the excessive excision activity in our reactions. The main type of translocation hurdle that DNAP would encounter on M13 ssDNA, or in general during lagging strand synthesis, is secondary structures in the template DNA (Fig 5G). The presence of gp2.5 single-stranded DNA binding protein in our reactions should alleviate some of the problems from secondary structures in template DNA, but perhaps not all and unusually stable hairpin-type structures would require strand displacement synthesis.

If secondary structures are responsible for the excessive excision activity, then we expect a lower Pol/Exo ratio in the absence of gp2.5, conditions under which T7 DNAP would be facing secondary structures in the template more frequently, in comparison with reactions with gp2.5. We carried out primer extension reactions on M13 ssDNA in the absence of gp2.5. Indeed, the Pol/Exo ratio in the absence of gp2.5 was $2.1 \pm 0.1$ in comparison with $33 \pm 3$ in the presence of gp2.5 (Fig 5H and Appendix Fig S2). These results indicate that secondary structures in template DNA are responsible for the majority of correct nucleotide excision during DNA synthesis on M13 ssDNA. Based on T7 gp2.5 footprint of ~7 nucleotides on ssDNA (Hernandez & Richardson, 2019), the gp2.5 concentrations were sufficient to saturate the M13 ssDNA (Table 2). Doubling the concentration of gp2.5 protein did not change the Pol- and Exo-activities (Fig 5H and Appendix Fig S2).

## Slowed helicase and helicase–polymerase uncoupling causes excessive excision of nucleotides

Translocation problems from secondary structures in DNA can explain excessive excision of correctly incorporated nucleotides during lagging strand synthesis. However, we also observed a similarly high excision activity during leading strand synthesis. To understand the basis for the excision activity during leading strand synthesis, we fit the Pol-activity and Exo-activity trends from the mini-circle leading strand synthesis assay collected at various dVTP concentrations (Fig 1C and D) to the spontaneous partitioning model and Pol-inhibited model. The spontaneous partitioning model predicts a sharp decrease in the Exo-activity at high dNTP levels, whereas the data show no reduction in Exo-activity even at high dVTP concentrations of 300 μM and dTTP at 500 μM (Fig 6A and B). Hence, kinetic modeling suggests that excessive excision activity during leading strand synthesis is caused by translocation problems as well.

What type of translocation problems occurs during leading strand synthesis? We have shown that T7 DNAP can unwind up to two base-pairs of the duplex DNA at the fork junction (Nandakumar *et al*, 2015). However, DNA synthesis stops after two base-pair synthesis and extended synthesis through the duplex DNA requires the presence of T7 helicase. A coupled T7 DNAP and T7 helicase complex, which is optimally unwinding and synthesizing DNA, can efficiently trap the newly unwound nucleotides after DNA synthesis and prevent the newly unwound bases from reannealing (Nandakumar *et al*, 2015). However, if the helicase becomes uncoupled from the DNAP, the unwinding rate slows down. and under these conditions, DNA reannealing may occur and promote DNAP backtracking and consequently primer-end fraying (Fig 6C). Since the frayed

primer-end resembles a mismatched state, it will be excised rapidly. Thus, similar to secondary structures in lagging strand template, the replication fork itself is a translocation hurdle during leading strand synthesis. Both obstacles promote reannealing of newly unwound bases to create frayed primer-ends that are easily partitioned in the Exo-site.

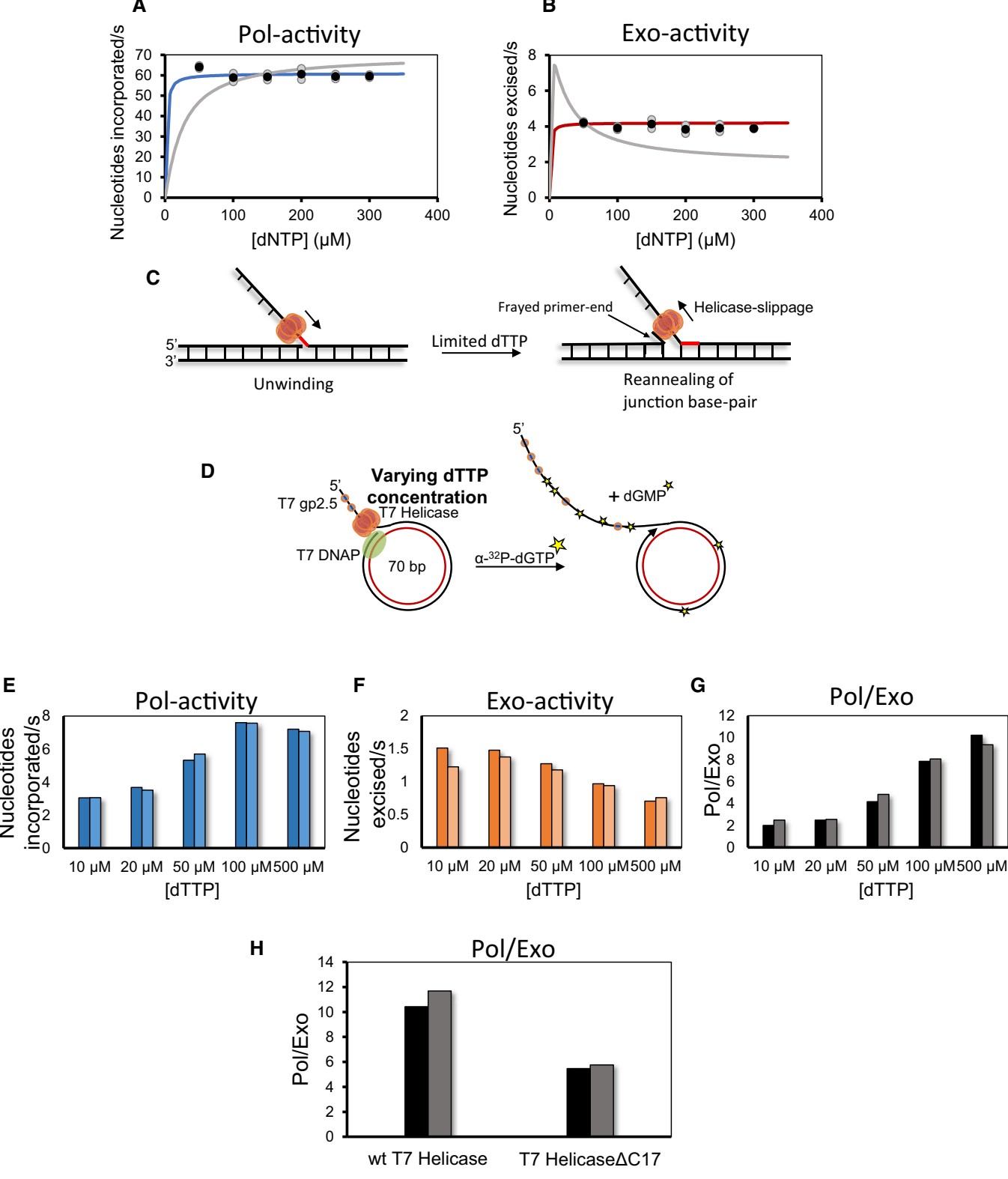

**Figure 6.**

◀

**Figure 6.  Slowed helicase or helicase–polymerase uncoupling causes excessive excision of nucleotides.**

A, B   Nucleotides incorporated (A) and excised (B) per second (black circles) determined from the slopes of linear fits of data in Fig 1C and D. Blue- and red-colored lines are trends from the global fitting of Pol- and Exo-data to the complete model shown in Figs 7 and EV5. Gray curve represents the poor fit of the data to the model in the blue box in Figs 7 and EV5. Gray circles are data points from two individual experiments. Black circles show the mean data that were fitted to the models.

C       Model depicting reannealing of the newly unwound base-pair (in red) at a replication fork and consequently primer-end fraying.

D       The experimental design to measure the effect of increasing dTTP concentration on Pol- and Exo-activities.

E, F    Moles of nucleotides incorporated (E) and excised (F) per mole of substrate DNA per second during leading strand synthesis reactions performed with 10 μM dVTPs and different dTTP concentrations.

G       Pol/Exo ratios for leading strand synthesis reactions performed with varying levels of dTTP as determined from (E and F).

H       Pol/Exo ratio during leading strand synthesis reactions performed with wt T7 helicase and its C-tail deletion mutant (T7 HelicaseΔC17). Reactions were performed with 150 μM dVTPs and 500 μM dTTP.

Data information: Bars show the data from two individual experiments.

To modulate the degree of uncoupling, we slowed down the helicase. It is known that dTTP binding hydrolysis drives the stepping mechanism of T7 helicase (Liao *et al*, 2005; Donmez & Patel, 2008; Pandey & Patel, 2014; Syed *et al*, 2014). We can, therefore, control helicase's translocation and unwinding rate by varying the concentration of dTTP. We expect that under sub-saturating dTTP conditions when helicase is not moving efficiently, fork regression and Exo-activity will be higher. In contrast, at saturating dTTP conditions when the helicase is unwinding the duplexed DNA optimally, the Exo-activity will be lower.

Although the preferred fuel for T7 helicase is dTTP, it can utilize other nucleotides, in particular dATP and dGTP (Pandey & Patel, 2014). Therefore, we lowered the dVTP concentration to 10 μM and varied the dTTP concentration from 10 to 500 μM (Fig 6D and Appendix Fig S3A). At 10 μM dTTP conditions, the Pol/Exo ratio was close to two (Fig 6G). This Pol/Exo ratio represents an extreme case of translocation hurdle that removes most of the incorporated nucleotide resulting in DNAP idling. This type of idling is beneficial for creating ligatable nicks when DNAP encounters the previous Okazaki fragment (Garg *et al*, 2004). With increasing dTTP, Pol-activity increased (Fig 6E and Appendix Fig S3B) both due to the availability of dTTP for DNA synthesis and due to the increased rate of helicase translocation. Consistent with our hypothesis, the Exo-activity was higher at sub-saturating dTTP concentration and become lower as we increased dTTP levels (Fig 6F and Appendix Fig S3C). Therefore, the overall Pol/Exo ratio increased with higher dTTP concentrations (Fig 6G). These experiments demonstrate that a consequence of slowing the helicase during leading strand synthesis is excision of correctly incorporated nucleotides. We observed the hydrolysis of dGTP to dGDP in our reactions, and as expected, the rate of dGTP hydrolysis decreased with dTTP concentration (Appendix Fig S3D).

Another reason for polymerase backtracking and primer-end fraying is helicase–polymerase uncoupling. It is known that T7 helicase and T7 DNAP are physically coupled via the C-terminal tail of T7 helicase (Notarnicola *et al*, 1997). Leading strand synthesis reactions with T7 DNAP and T7 helicase ΔC17 showed a Pol/Exo ratio of $5.7 \pm 0.4$, which is 50–60% lower than with the WT T7 helicase (Fig 6H). The lower Pol/Exo ratio indicates that occasional helicase–polymerase uncoupling can also result in relatively large excision of the correctly incorporated nucleotides during leading strand DNA synthesis. Removal of ~7% of the correctly incorporated nucleotides on templates with no apparent lesions suggests that these events are not rare and contribute in a major way to excision of nascent DNA during normal DNA synthesis.

## Discussion

### Excessive exonuclease activity of correct nucleotides during DNA synthesis

T7 DNAP is a high-fidelity replicative polymerase that makes one mistake for every $10^5$–$10^6$ correct nucleotide incorporation events. It also has a proofreading exonuclease activity that corrects this mistake efficiently. Theoretically, if nucleotide excision takes place strictly after misincorporation, we expect a Pol/Exo ratio of $10^5$–$10^6$ or almost undetectable excision activity during DNA synthesis. However, our study shows that T7 DNAP excises 6–8% of incorporated nucleotides during leading and lagging strand syntheses, which strongly suggests that correctly incorporated nucleotides are removed during DNA synthesis. The presence of a protein trap did not change the Pol/Exo ratio, which indicates that excessive excision is due to primer-end partitioning from the Pol-site to Exo-site during processive DNA synthesis. Moreover, this phenomenon is not peculiar to T7 DNAP, as we observed similarly high excision rates during DNA synthesis by other replicative DNAPs, such as the yeast mitochondrial DNAP and bacteriophage Φ29 DNAP, which belong to A-family and B-family of polymerases, respectively. Thus, excessive excision of correctly incorporated nucleotides during processive DNA synthesis is a general feature of replicative DNAPs.

Interestingly, excessive excision during DNA synthesis by *E. coli* Pol I was reported almost four decades ago by Alan Fersht (Fersht *et al*, 1982); however, since then, it has not been studied in detail. Fersht speculated that excessive excision occurred from spontaneous partitioning of the correctly base-paired primer-end into the Exo-site and it is the "cost of proofreading". Several optical tweezer-based single-molecule studies have reported excessive exonuclease activity of DNAPs more recently (Wuite *et al*, 2000; Ibarra *et al*, 2009; Hoekstra *et al*, 2017). However, these studies observed excision of several hundreds of nucleotides at a time between synthesis events. A caveat of these force-based experiments is that polymerase and exonuclease activities are measured by applying tension to the primer–template junction that stretches the template DNA and creates a scenario equivalent to an incorrectly incorporated nucleotide. This is consistent with the observation that the probability of primer-end binding to the exonuclease site was proportional to the tension (Hoekstra *et al*, 2017). Moreover, the constant tension on the template created a scenario akin to a continuous series of misincorporated nucleotides, which explains why the primer-end remains at the Exo-site for an extended period, and there is processive excision of a large

number of nucleotides. Given these limitations of the force-based experiments, the origin of excessive removal of correctly incorporated nucleotides during DNA synthesis remained unclear. Since we happened to observe excessive excision in three replicative DNAPs, we carried out a detailed study to investigate its mechanistic basis using the model T7 DNAP.

First, we show that excessive excision does not arise from processive degradation of the nascent DNA as suggested by single-molecule experiments discussed above. We introduced phosphorothioate linkages as molecular brakes to inhibit processive excision of the nascent DNA and observed that such linkages did not affect the Pol/Exo ratio. Thus, the only reasonable explanation was that excessive excision of correctly incorporated nucleotides results from frequent intramolecular shuttling of the correctly base-paired primer-end between Pol-site and Exo-site during ongoing DNA synthesis. Primer-end shuttling between Exo-site and Pol-site has been observed in single-molecule FRET experiments with the Klenow fragment of polymerase I (Lamichhane et al, 2013) and nanopore experiments (Lieberman et al, 2014), but such shuttling of the primer-end during active DNA synthesis has not been measured.

To understand the mechanistic basis of frequent shuttling of the correctly base-paired primer-end between the two active sites, we examined the link between translocation and excision. Using a nanopore-based method, Lieberman et al showed that in a non-replicating Φ29 DNAP (absence of DNA synthesis), the primer-end binds to the Exo-site from the pre-translocation state (Lieberman et al, 2014). Our kinetic studies are consistent with this model. First, we show that on a given DNA sequence, there is a direct correlation between pyrophosphorolysis, which occurs from the pre-translocation state, and the excision rate of the correct nucleotides. Second, we show that an abasic site-modified template that destabilizes the pre-translocation state also decreases the exonuclease activity. Thus, our experiments are consistent with the model that primer-end binding to the Exo-site occurs from the pre-translocation state of the DNAP.

Additionally, the link between the DNAP translocation state and primer-end transfer to the Exo-site is supported by studies of the antimutator strains of bacteriophage T4 DNAP (Reha-Krantz et al, 1993; Reha-Krantz, 1995). A well-studied antimutator, T4 DNAP with mutation A737V, exhibits sensitivity to phosphonoacetic acid, an analog of PPi (Reha-Krantz et al, 1993). This mutant shows excessive excision of correct nucleotides in comparison with wild-type T4 DNAP (Muzyczka et al, 1972) and exhibits difficulty in translocation, particularly on a DNA substrate which requires strand displacement (Gillin & Nossal, 1976).

**Excessive excision of correctly incorporated nucleotides arises from translocation problems**

If the source of excessive excision was because the correctly base-paired primer-end shuttles to the Exo-site spontaneously and frequently during normal nucleotide incorporation cycles then increasing dNTP levels should decrease the Exo-activity. We verified this prediction by kinetic modeling where we modeled DNA synthesis using repetitive nucleotide incorporation cycles (Fig 7, blue box and Fig EV5). We used the measured rate constants for each of the elementary steps in the nucleotide incorporation cycle and assumed that translocation was a rapid equilibrium step, as suggested by

nanopore experiments (Lieberman et al, 2014). This spontaneous partitioning model (Fig 7, blue box) predicted that the Exo-activity should decrease with increasing dNTP concentrations (Figs 5E and 6B, gray lines). However, our measurements show that the Exo-activity remains constant and does not reduce with increasing dNTP levels (Figs 5E and 6B). We, therefore, concluded that excessive excision activity is not due to spontaneous shuttling of the primer-end into the Exo-site during normal DNA synthesis. In other words, it is not the "cost of proofreading" as suggested previously (Fersht et al, 1982).

We wondered if excessive excision activity could arise from translocation problems during DNA synthesis, wherein DNA synthesis would be paused and this would increase the chance of primer-end partitioning into the Exo-site. We modeled such a scenario of occasional translocation problems by creating Pol-inhibited states in a branched pathway from the intermediate states in the normal DNA synthesis pathway. The Pol-inhibited states are incompetent in primer extension, but competent for active-site shuttling and excising the primer-end (Figs 7 and EV5). This model accurately predicted the observed unchanged excision rates as a function of dNTP concentrations both during synthesis on the single-stranded M13 template and the minicircle template (Figs 5E and 6B).

Pol-inhibited states during DNA synthesis can be generated by various types of replication hurdles (Mirkin & Mirkin, 2007). Secondary structures in DNA such as hairpin, G-quadruplex, cruciform, and triplex and template lesions create Pol-inhibited states, where the primer-ends are not readily extendible. These unstable primer-ends are responsible for genome instability, but they could also be easily shuttled into the Exo-site for excision. Excision and incorporation over lesions can occur in a repetitive manner to result in excessive excision of the newly incorporated nucleotides at these sites due to polymerase idling (Khare & Eckert, 2002; Meng et al, 2009). Thus, one of the consequences of encountering such natural replication hurdles during DNA synthesis is excision of the correctly incorporated nucleotides.

The M13 ssDNA template used in our assay contains many stable DNA secondary structures (Reckmann et al, 1985) that can also generate these Pol-inhibited states. Most DNAPs require single-stranded DNA binding protein to catalyze efficient DNA synthesis through secondary structures; however, the single-stranded binding protein cannot fully resolve stable secondary structures which become problematic for DNAP translocation. Our studies show that the source of excessive correct nucleotide excision on the M13 ssDNA template is such stable secondary DNA structures because leaving out T7 gp2.5, the single-stranded binding protein, resulted in a substantial decrease in the Pol/Exo ratio. T7 DNAP can unwind and incorporate a few nucleotides through stable secondary structures such as hairpins, but the hairpin structure can reanneal, which would lead to polymerase backtracking and creation of a frayed primer-end, which akin to a mismatch is efficiently shuttled into the Exo-site for excision.

We propose that the same phenomena of backtracking and primer-end fraying lead to excessive excision of correct nucleotides during leading strand synthesis. The mechanism of DNAP backtracking during leading strand synthesis, however, is somewhat different. Rather than stable secondary structures, the replication fork itself presents a barrier. We have shown that T7 DNAP can unwind a few base-pairs at the replication fork junction, but requires T7 helicase to trap the unwound bases by unidirectional forward

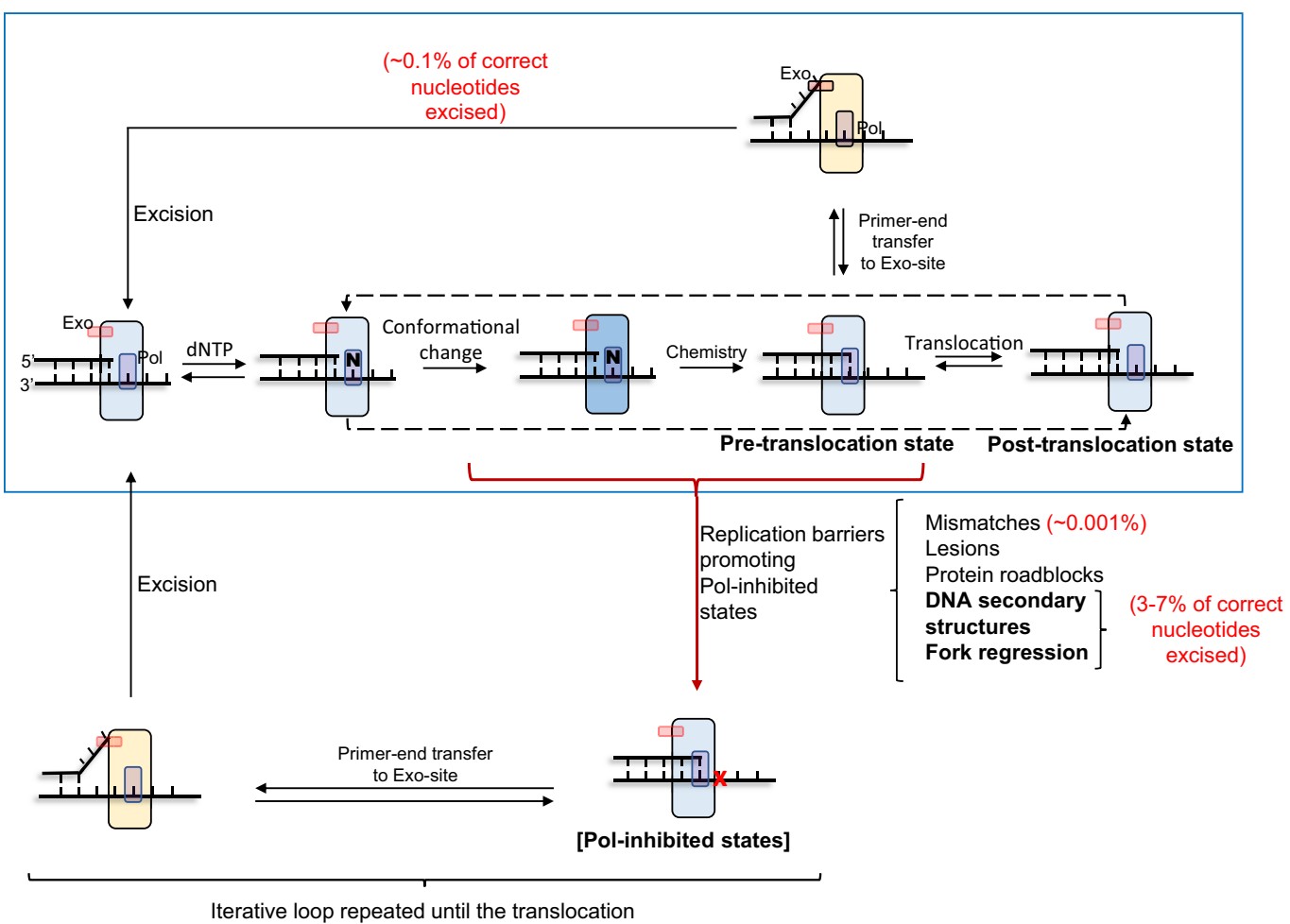

**Figure 7. Translocation-regulated model of active-site switching.**

The DNA synthesis pathway in the blue box is the canonical pathway of nucleotide incorporation cycles. Nucleotide binding to the post-translocation state of DNAP is followed by fingers closing conformational change and the chemical step of nucleotide incorporation that result in the extension of the primer-end by one nucleotide. Upon PPi dissociation, the DNAP ends up in the pre-translocated state (fingers open). The primer-end in the pre-translocated state can partition into the exonuclease active site, or the DNAP can translocate to the post-translocated state to repeat the next cycle of nucleotide incorporation. We assumed here that the pre- to post-translocation step is a rapid equilibrium step. Occasionally, the DNAP encounters translocation hurdles that create Pol-inhibited states. These are branched out of the normal DNA synthesis pathway and are not competent in translocation or dNTP binding but competent in shuttling the primer-end in the Exo-site and excising the primer-end nucleotide. After excision of the primer-end, the DNAP is shown to return to the post-translocation state. While DNA synthesis resumes via the normal pathway after excision event due to misincorporation, the DNAP idles in case of template lesions and other replication barriers until the problem is resolved.

translocation (Nandakumar *et al*, 2015). Due to occasional helicase slippage or slowed helicase rate, the DNA synthesis and unwinding events become uncoupled, resulting in DNAP backtracking and fraying of the primer-end. We tested this model in several ways by showing that the excision activity increases as T7 helicase is slowed, and the Pol/Exo decreases when we use a T7 helicase mutant that would be prone to uncoupling because it cannot physically interact with the leading strand T7 DNAP.

We observed that the Pol/Exo ratio during lagging strand synthesis reactions on minicircle DNA substrate is around 16, which is lower than the Pol/Exo ratio of ~35 from primer extension reactions on primed M13 ssDNA. Similar to the conclusions drawn from the M13 ssDNA experiments, the source of excessive excision during lagging strand synthesis could be the presence of DNA secondary structures in the lagging strand template (Fig EV1Biii). Furthermore, there is additional source of potential excision activity when the lagging strand DNAP completes the synthesis of an Okazaki fragment, encounters the 5′-end of the previous Okazaki fragment, and remains bound to the primer-end. Under these conditions, the DNAP can idle between Pol- and Exo-sites until it recycles to a newly synthesized primer (Garg *et al*, 2004). We speculate that DNAP idling in the time window between two Okazaki fragment syntheses contributes to excessive excision during lagging strand synthesis. Additionally, as a component of the replisome, the lagging strand DNAP interacts with the N-terminal domain of T7 helicase–primase (Gao *et al*, 2019). Such interactions, especially those involving the exonuclease domain of DNAP, could alter the dynamics of active-site switching.

Does excessive excision of the correctly incorporated nucleotides have any mechanistic advantage in terms of improving the accuracy of DNA synthesis? It is known that exonuclease-deficient polymerase undergoes mutagenic synthesis over lesions which is undesirable, whereas the exonuclease proficient polymerase idles at the lesion sites and slows down or prevents such mutagenic synthesis (Meng *et al*, 2009; Vaisman & Woodgate, 2017; Marians, 2018). Interestingly, two recent studies suggested that primer-end shuttling to the Exo-site is a protective mechanism to prevent mutagenic DNA synthesis over natural replication impediments (Parkash *et al*, 2019; Xing *et al*, 2019). These studies showed that a tumorigenic Pol ε mutation that interferes with primer-end binding to the Exo-site has an increased mutagenic synthesis rate over secondary structures in DNA and mismatched primer-end. They suggested that the defect in primer-end binding to the Exo-site confers the unusual tumorigenic activity. Earlier studies of T4 DNAP mutants had also identified several mutants with a strong mutator phenotype that were defective in primer-end binding to the Exo-site (Stocki *et al*, 1995). Thus, primer-end binding to the Exo-site at replication barriers seems to be an important mechanism to protect the primer-end from aberrant extensions. Therefore, partitioning of the primer-end between the polymerase and exonuclease sites not only proofreads misincorporations but also protects the primer-end from mutagenic extensions at replication barriers.

Interestingly, the energetic cost of protecting the primer-end from mutagenic extension is much greater than that of proofreading misincorporations. It is possible that there are additional cellular factors that resolve these replication hurdles and reduce the energetic cost of primer-end protection. In contrast to T7 replisome, eukaryotic replisomes are complex and constitute of many protein factors (O'Donnell *et al*, 2013; Pellegrini & Costa, 2016). There have been successes in reconstituting the eukaryotic replisome *in vitro* (Yeeles *et al*, 2017; Taylor & Yeeles, 2018). Hence, the extent of excision during DNA synthesis by the eukaryotic replisome and the effect of the individual components on the proofreading cost can be studied in the future.

# Materials and Methods

### Nucleotides

Nucleotides were purchased either from Promega (Madison, WI) or TriLink BioTechnologies (San Diego, CA). Radiolabeled deoxynucleotides were purchased from Perkin Elmer (Waltham, MA).

### Oligonucleotides

All oligonucleotides were purchased from Integrated DNA Technologies (Coralville, IA) with PAGE or HPLC purification. Sequences of the DNA substrates used for the study are tabulated in Table 1.

### Proteins

T7 gp4A′ (M64L mutation to prevent translation of gp4B), C-tail mutant of T7 gp4A′ (T7 HelicaseΔC17), gp5 (exo+), and gp5 (exo−) proteins were purified as described previously (Donlin *et al*, 1991; Pandey *et al*, 2009). T7 gp2.5 protein was purified as described

previously (Kim *et al*, 1992). Mip1 and Rim1 were purified as described earlier (Ramanagoudr-Bhojappa *et al*, 2013; Ramachandran *et al*, 2016). *Escherichia coli* Thioredoxin was purchased from Sigma-Aldrich (St. Louis, MO). Apyrase and Φ29 DNAP were purchased from New England Biolabs (Ipswich, MA).

### Measurement of excision events relative to polymerization events

Polymerase-to-exonuclease ratio for T7 DNAP was measured at 18°C on a 70-nt minicircle template DNA annealed to a 110-nt primer. The minicircle-primer substrate had a 40-nt 5′ overhang to assemble T7 helicase (gp4A′). For experiments with T7 DNAP, the minicircle DNA substrate (32 nM) was incubated with gp4A′ (400 nM), dTTP (final concentrations as mentioned in Figures and/or Figure legends), and 1.5 mM EDTA for 30 min after which 400 nM T7 DNAP (complex of T7 gp5 and *Escherichia coli* thioredoxin) was added (Mixture A). Mixture B consisted of T7 gene 2.5 protein (gp2.5) (20 μM), dVTPs (dATP, dCTP, and dGTP, final concentrations mentioned in Figures/Figure legends), 10 mM MgCl$_2$, and small amount of radiolabeled nucleotide, α-$^{32}$P-dGTP. Reactions were initiated by mixing equal volumes of Mixtures A and B. Reactions were quenched at different time points by adding equal volumes of 8 M formic acid. For background correction, formic acid was added to Mixture A before the addition of Mixture B. 1 μl of each of the quenched reactions was spotted on PEI Cellulose F TLC sheets (Millipore-Sigma). Analytes were separated by TLC (polyethyleneimine-cellulose) with potassium phosphate buffer (pH 3.8). Sheets were air-dried and were used to expose the phosphor-imaging screens for a few hours. Exposed screens were scanned using Typhoon FLA 9500 imaging system (GE Healthcare). Spot intensities for dGMPs incorporated in the extended primer DNA, unutilized dGTP, dGDP, and dGMP excised for reactions quenched at different time points were quantified and corrected for the background. Amounts of dNMP incorporated and excised were calculated from the quantified intensities and normalized for the quantity of DNA substrate and concentration of dVTPs (moles of nucleotides incorporated or excised per mole of the replisome complex). Data were plotted against time and were fitted to linear trends to obtain slopes (nucleotides incorporated or excised per mole of the replisome complex per second). We assumed the amount of the replisome complex to be same as the amount of minicircle substrate DNA used in the assay as it was a limiting reagent in comparison with protein concentrations used. The slopes were used to calculate the Pol/Exo ratios.

Variations of the above experimental strategy were employed to determine dTTP concentration dependence (at 10 μM dVTPs and different concentrations of dTTP ranging from 10 to 500 μM), dVTP concentration dependence (experiments performed at dTTP concentration fixed at 500 μM and with varying concentrations of dVTPs), and the effect of the presence of helicase deletion mutant. Similar experiments were performed with exonuclease-deficient mutant of T7 DNAP (exo- T7 gp5 D5A, E7A). To study the effect of α-thiodNTPs (dNTPαS) on Pol/Exo ratio, we included dTTPαS, dCTPαS, or dATPαS in Mixture B.

Experiments to determine the Pol/Exo ratio for bacteriophage Φ29 DNAP were performed at 30°C in the absence of a single-stranded DNA binding protein or helicase. Final concentration of Φ29 DNAP used in reactions was 1 μM. Pol/Exo ratio for yeast

**Table 1.  Oligonucleotide sequences used in the study.**

| Oligo | Figure | Sequence |
|---|---|---|
| 70 nucleotide minicircle DNA | Figs 1–3 and 6, and Appendix Figs S1, S2 and S5 | 5′ CACCATATCCTCGACCATCCCCAATATGGTCCATCAACCCTTCACCTCACTTCACTCCA CTATACC ACTC 3′ |
| Primer for the 70 nucleotide minicircle DNA | Figs 1–3 and 6, and Appendix Figs S1, S2 and S5 | 5′ TTTTTTTTTTTTTTTTTTTTTTTTTTTTTTTTTTTTTTTTTTGAGTGGTATAGTGGAGTGAAGTG AGGTGAAGGGTTGATGGACCATATTGGGGATGGTCGAGGATATGGTG 3′ |
| Primer 6236 for M13 ssDNA | Fig 5 and Appendix Fig S1F | 5′ GACTCTAGAGGATCCCCGGGTACCGAGCTC 3′ |
| Matched primer 25-mer | Appendix Fig S4 | 5′ 6-FAM/TGGTTAGTGGAAGAGATTCACAAAC 3′ |
| Template 55-mer | Appendix Fig S4 | 5′ ACTTATCACACATTCTTCTCATTAACTTGTGTTTGTGAATCTCTTCCACTAACCA 3′ |
| Matched Primer 35-mer | Fig 4 and Appendix Fig S3 | 5′ FAM/AGGAGTGCGCTGGTTAGTGGAAGAGATTCACAAAC 3′ |
| Template 65-mer | Fig 4 and Appendix Fig S3 | 5′ ACTTATCACACATTCTTCTCATTAACTTGTGTTTGTGAATCTCTTCCACTAACCAG CGCACTCCT 3′ |
| Abasic Template 55-mer | Appendix Fig S4C | 5′ ACTTATCACACATTCTTCTCATTAACTTG/Abasic/GTTTGTGAATCTCTTCCACTAACCA 3′ |
| Mismatched Primer 35-mer | Fig 4 and Appendix Fig S3 | 5′ FAM/AGGAGTGCGCTGGTTAGTGGAAGAGATTCACAAAG 3′ |
| Mismatched Primer 35-mer with Phosphorothioate linkage | Appendix Fig S2A | 5′ FAM/AGGAGTGCGCTGGTTAGTGGAAGAGATTCACAAA*G 3′ |
| | | *Phosphorothioate linkage |
| Trap 1 | Fig 4 and Appendix Figs S2 and S3 | 5′ AGGAGTGCGCTGGTTAGTGGAAGAGATTCACAAAC 3′ |
| Matched Primer 35-mer with 3′dideoxy end | Fig 1F and Appendix Fig S1C | 5′ AGGAGTGCGCAGGAAAGAGGAAGAGAAACACAAAddC 3′ |
| Matched template 68-mer | Fig 1F and Appendix Fig S1C | 5′ CATTATCACACATTCGTCTCATTGACTTGTCTGGTTTGTGTTTCTCTTCCTCTTTCC TGCGCACTCCT 3′ |
| Fork DNA 1 | Appendix Fig S2Bi and Bii | 5′ BHQ1/GACATGACTGACGAGAGAGTCTTGTGATGCTCCTACGTAGGCTACGCTATGTCGTCA AGTTCACACTGTTACTAG 3′ |
| Fork DNA 2 | Appendix Fig S2Bi and Bii | 5′ TTTTTTTTTTTTTTTTTTTTTTTTTTTTTTTTTTTTCTACGTAGGAGCATCACAAGACT CTCTCGTCAG TCATGTC/FAM 3′ |

mitochondrial DNA polymerase, Mip1, was similarly determined in the absence of a helicase. Reactions were conducted at 30°C, and mitochondrial single-stranded binding protein, Rim1, was used. Mip1 and Rim1 final concentrations in the reaction mixture were 200 nM and 16 μM, respectively.

## Measurement of Pol/Exo ratios during lagging strand synthesis and coupled leading strand synthesis

Incorporation and excision of the nucleotides during lagging strand synthesis were determined using 70 bp minicircle DNA substrate. The assays were conducted in the presence of T7 DNA helicase–primase and T7 DNAP. 500 μM each of the priming ribonucleotides, ATP and CTP, was added to Mixture A. To measure Pol/Exo ratio during lagging strand synthesis, Mixture B was spiked with $\alpha$-$^{32}$P-dCTP. Pol/Exo ratio for leading strand synthesis coupled to lagging strand synthesis reactions was measured with $\alpha$-$^{32}$P-dGTP in the presence of ATP and CTP.

## Dependence of polymerase and exonuclease activities on dNTP concentration

dNTP concentration dependence experiments were performed on M13 mp18 ssDNA annealed to a 30-nucleotide primer. The concentration of primed M13 ssDNA substrates used for each dNTP

concentration is provided in Table 2 (1:10 ratio of M13 ssDNA and the 30-mer primer was used in annealing reaction). DNA substrate was incubated with T7 gp2.5 (concentrations used are provided in Table 2), 50 mM Tris–HCl (pH 7.5), 40 mM NaCl, 10% glycerol, and 1.5 mM EDTA for 30 min. 200 nM DNAP was added to make Mixture A. Mixture B constituted of the four dNTPs, 10 mM MgCl$_2$, 50 mM Tris–HCl (pH 7.5), 40 mM NaCl, and 10% glycerol and was spiked with $\alpha$-$^{32}$P-dGTP. All the reactions were performed at 18°C. Reactions were initiated by mixing equal volumes of Mixtures A and B. Reactions were quenched at different time points by addition of 4 M final concentration of formic acid. Reactions were spotted on TLC sheets, and the data for fraction incorporated and fraction excised at different dNTP concentrations were obtained using the method described above. Incorporation and excision slopes and Pol/Exo ratio were plotted against dNTP concentrations used. Incorporation data were fitted in hyperbolic trend to obtain Km for dNTP dependence.

## Pre-steady-state kinetics of excision on short primer–template

Two sets of matched primer–template were used for the experiments (25-mer primer annealed to the 55-mer template and 35-mer primer annealed to the 65-mer template). Primers had fluorescein labels at 5′ end. Pre-steady-state excision kinetics for the annealed DNA substrates were determined at 18°C using millisecond time

**Table 2.** M13 ssDNA and T7 gp2.5 concentrations used for dNTP concentration dependence experiments.

| dNTP concentration (µM) | M13 ssDNA concentration (nM) | T7 gp2.5 concentration (µM) |
|---|---|---|
| 1 | 2.5 | 2.5 |
| 2 | 2.5 | 2.5 |
| 5 | 2.5 | 2.5 |
| 10 | 8 | 8 |
| 15 | 8 | 8 |
| 20 | 8 | 8 |
| 25 | 8 | 8 |
| 30 | 8 | 8 or 16 |

range quenched-flow apparatus (Kintek). 200 nM annealed DNA substrate was incubated with 300 nM T7 DNA polymerase, 1.5 mM EDTA, 50 mM Tris–HCl pH 7.5, 2 mM DTT, 40 mM NaCl, and 10% glycerol (Mixture A). Mixture B consisted of 10 mM MgCl$_2$, 50 mM Tris–HCl pH 7.5, 2 mM DTT, 40 mM NaCl, and 10% glycerol. Both the mixtures were transferred to separate quenched-flow syringes. Reactions were initiated by mixing equal volumes of Mixtures A and B and were quenched at different time points (ranging from milliseconds to seconds range) by addition of the EDTA quench (150 mM final concentration). The zero-second reaction did not have MgCl$_2$ in Mixture B. Reactions were mixed with five times excess of Trap DNA 1 and sequencing dye (98% formamide, 10 mM EDTA, and 0.025% Bromophenol Blue), boiled at 95°C for 5 min, and immediately transferred to ice. Processed samples were loaded on to 24% polyacrylamide/6M urea sequencing gels. Gels were run overnight at constant power and were scanned on Typhoon FLA 9500 (GE Healthcare). Individual bands were quantified using ImageQuant TL software and were corrected for background. Excision rates were determined by fitting the data (using Kintek Explorer package) to a model with irreversible and sequential (one nucleotide at a time) excision of nucleotides on the 3′ end of the primer. Variations of the similar experiment with a mismatched 35-mer/65-mer primer/template, a mismatched 35-mer/65-mer primer/template with the primer having a phosphorothioate linkage at the 3′ end, and a matched 25-mer/55-mer primer/template with the template containing an abasic moiety at n+1 position (position next to the primer–template junction) were also carried out.

**Pre-steady-state kinetics of pyrophosphorolysis on short primer–template**

A 35-mer primer (with fluorescein label at 5′ end) annealed to a 65-mer template was used to determine the transient state kinetics of pyrophosphorolysis reactions performed at 18°C. 200 nM DNA substrate was incubated with 300 nM exonuclease-deficient mutant of T7 DNAP (exo− T7 gp5 D5A, E7A), 20 mM sodium pyrophosphate, 1.5 mM EDTA, 50 mM Tris–HCl pH 7.5, 2 mM DTT, 40 mM NaCl, and 10% glycerol (Mixture A). Mixture B consisted of 10 mM MgCl$_2$, 15 units of Apyrase, 50 mM Tris–HCl pH 7.5, 2 mM DTT, 40 mM NaCl, and 10% glycerol. Reaction buffers A and B were

transferred to separate syringes fitted to a quenched-flow apparatus (Kintek). Reactions were initiated by mixing equal volumes of Mixtures A and B and were quenched at different time points (ranging from milliseconds to seconds) with rapidly mixing EDTA quench (150 mM final concentration). For a zero-second reaction, Mixture B did not contain MgCl$_2$. Treated reactions were loaded on to 24% polyacrylamide/6 M urea gel, and rates of pyrophosphorolysis reactions were determined as mentioned above.

Similar pyrophosphorolysis reactions conducted on a matched 25-mer/55-mer primer/template and a matched 25-mer/55-mer primer/template with the template containing an abasic moiety at n+1 position (position next to the primer–template junction) were quenched at 15 and 30 s by hand. Treated reactions were processed as mentioned above and were loaded on to a 24% polyacrylamide/6 M urea gel. Band intensities were quantitated to determine the comparative extent of pyrophosphorolysis.

**DNA unwinding assay**

The T7 helicase unwinding assays were performed at 18°C in rapid stopped-flow apparatus (Kintek). Mixture A (50 mM Tris–HCl, pH 7.5, 40 mM NaCl, 10% Glycerol, 2 mM DTT, 1.5 mM EDTA, 20 nM fork DNA substrate, 1 mM dTTP or dTTPαS, and 100 nM T7 helicase hexamer) and Mixture B (50 mM Tris–HCl, pH 7.5, 40 mM NaCl, 10% glycerol, 2 mM DTT, and 10 mM MgCl$_2$) were taken in separate syringes A and B, respectively, and were mixed to initiate the reactions. Fluorescence intensity was measured in real time with excitation at 480 nm and emission monitored at 515 nm. A fork DNA substrate (prepared by annealing fork DNA 1 with a BHQ 1 moiety at its 5′-end and fork DNA 2 with a 3′-fluorescein moiety) which when annealed has the fluorescein fluorescence quenched by BHQ 1. Unwinding by T7 helicase results in the increase in measured fluorescence intensity which was used to determine rate of unwinding.

**Expanded View** for this article is available online.

## Author contributions

AS and MP designed the study, generated, and analyzed the data. MP purified T7 proteins. AS and DN performed kinetic modeling. YWY provided purified Mip1 protein. KDR provided purified Rim1 protein. AS and SSP wrote the manuscript. All authors approved the final version of the manuscript.

## Conflict of interest

The authors declare that they have no conflict of interest.

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
