## [Review Process File · The EMBO Journal]

Excessive excision of correct nucleotides during DNA synthesis explained by replication hurdles

Anupam Singh, Manjula Pandey, Divya Nandakumar, Kevin D. Raney, Y. Whitney Yin and Smita S. Patel.

Review timeline:

Submission date:	2 nd September 2019
Editorial Decision:	17 th October 2019
Revision received:	14 th November 2019
Editorial Decision:	29 th November 2019
Revision received:	23 rd December 2019
Accepted:	7 th January 2020

Editor: Hartmut Vodermaier

Transaction Report:

1st Editorial Decision

17th October 2019

Thank you for submitting your manuscript on DNA polymerase excision of correct nucleotides for our editorial consideration. It has now been assessed by three expert referees, whose comments are copied below for your information. As you will see, all referees appreciate your findings, their presentation, as well as the quality of the data. We shall therefore be happy to consider the study further for EMBO Journal publication, pending adequate revision of a number of specific points raised by the reviewers.

Most of the referees' points pertain to clarifications and discussions, and since they are well-explained in the reports I will not repeat them here. Regarding referee 2's more general concern about the main novelty/advance, my own reading of the study did not leave me with the impression that previous results would have been downplayed or improper novelty claims made; a view that was also shared by referee 1, who pointed out that Pavlov et al 2004 presented similar findings only in one figure and without further discussing their significance or mechanistic basis. In this light, I would not consider major re-writing of the study necessary here.

REFeree REPORTS

Referee #1:

Here, the authors show that, depending on conditions, approx. 2-15% of dNMPs correctly incorporated by T7 DNAP are subsequently excised by "proofreading". They further demonstrate clearly that this activity requires a functional exonuclease site, and that two other replicative polymerases also show this high level of exonucleolysis during normal DNA synthesis. They then proceed to dissect this activity of T7 DNAP under various conditions of limiting dNTP concentrations, etc., to demonstrate conclusively that exonucleolysis is enhanced under conditions where the polymerase may be transiently stalled by replication roadblocks (DNAP inhibited state).

Ultimately, they conclusively derive a kinetic model that fits the trends in all of their data.

This is a beautiful, comprehensive, innovative and provocative study, one that challenges current dogma and will have a major impact on our field.

My only comments to be considered in revision are minor, but should be addressed:

1. Concentrations of some reagents used in particular experiments have been omitted from the Materials and Methods or figure legends. Examples: phi29 DNAP (top of page 21); 30-nucleotide primer (middle of page 21); gene 2.5 protein (middle page 21); trap in legend to Figure 1F; dNTPs in Figure 2A.
2. Page 21, typo 5th line from bottom: "formic".
3. Figure 1B: Reaction times should be indicated on the figure.
4. Figure S5D shows production of dGDP, presumably as a result of use of dGTP by the helicase at limiting dTTP concentrations (should be explained in the legend). Units are not explicitly given, but I assume they relate directly to Figures S5B and C. Figure S5D should be referred to briefly in the text and the origin of the dGTP and units on the y-axis should be clarified in the figure legend. Indeed, the authors should carefully check that all other figures (including those in SI) are referred to in the text.
5. Figures 7 and S6: "Nucleosomes, and other proteins" - nucleosomes are of course irrelevant in bacterial systems, and evidence that they impede replisomes in eukaryotic systems would need to be cited. I would suggest being more general by citing "Protein roadblocks".
6. Figure S3, title typo: exonucleolysis

Referee #2:

This is a manuscript with an elegant series of experiments exploring proofreading in the context of the T7 replisome. In the first key experiment the authors monitor the ratio between dNTPs and dNMPs using thin-layer chromatography. To their surprise they found excision of correct nucleotides. Then they follow up this observation by investigating the cause. In brief, they show that replication hurdles, secondary structures in the DNA, slower unwinding of DNA by the helicase, or uncoupled helicase-polymerase, generate frayed primer-ends that are shuttled to the exonuclease site and excised efficiently.

Major points

The concept of excessive excision of correct nucleotides has already been shown before and has, therefore, to me limited news value. The authors discuss Fersht, et al suggesting that in *E. coli* 7-15% of correct nucleotides are excised. In addition, it was shown by Pavlov, Maki, Maki, and Kunkel (BMC Biology 2004) that 22% of the correctly inserted nucleotides were removed as dTMP by a Family B polymerase in a minimal assay with only the DNA polymerase and a DNA template. The numbers are comparable since thin layer chromatography was used to quantify dTMP and dTTP after the reaction had proceeded for a while. Thus, figure 2 corroborates Pavlov and coworkers finding. Furthermore, the last sentence in the first paragraph on page 7 implying that this is shown for the first time in Family B polymerases needs to be rephrased. Page 7, "these experiments demonstrate that excessive excision of incorporated nucleotides is a general feature of replicative DNA polymerases.

I find the experiments interesting, but disagree with the theme of the paper claiming that excessive excision of correct nucleotides is a novel observation.

What I believe is novel is the approach taken in the investigation of the mechanism causing

excessive excision.

Specific points:

Page 3, "Based on the excision rates of correctly base-paired 3'-end nucleotide, this partitioning model predicts that only 0.1% of the correctly incorporated nucleotide will be excised during DNA synthesis". As, I perceive the manuscript, this is why the authors are surprised that the T7 DNA polymerase and other DNA polymerases perform excessive excision of correct nucleotides. Could it be that the model was incorrect and if so can you today clarify why that was the case?

Page 5, third paragraph, In all experiments are very high concentrations of deoxyribonucleotides used. In fact, they are much higher than the determined in vivo concentrations. Could the outcome of experiments exploring the effect of dNTP concentration been different if dNTP concentrations in the range 1-11 uM was used? Furthermore, would adding ribonucleotides at mM concentration also affect the balance between the exonuclease and polymerase activities? The dNTP concentration is a very important parameter in relation to the presented model on page 17; "This spontaneous partitioning model (Figure 7, blue-box) predicted that the Exo-activity should decrease with increasing dNTP concentrations (Figure 5E and 6B, gray lines). However, our measurements show that the Exo-activity remains constant and does not reduce with increasing dNTP levels (Figure 5E and 6B). We, therefore, concluded that excessive excision activity is not due to spontaneous shuttling of the primer-end into the Exo-site. In other words, it is not the "cost of proofreading" as suggested previously."

Perhaps the exonuclease activity decrease with increasing dNTP concentrations if the starting concentration is much lower. That was shown to be the case in figure 8 by Ganai et al, NAR 2015. How would measurements at much lower dNTP concentrations influence the final model that depends on the generation of frayed 3'-termini?

Page 5, last sentence on the page, "If the increased excisions were due to DNAP dissociating from the DNA substrate and rebinding of the primer-end preferentially in the Exo-site, the Pol/Exo ratio would decrease with the protein trap. The Pol/Exo ratio remained unchanged after the addition of the trap(Figure 1F, Figure S1C)." Would not the binding of DNAP to a primer with a 3'-ddNMP in the presence of the high dNTP concentration freeze the DNA in the polymerase site. Thus, I would expect a decrease in exonuclease activity and an increased Pol/Exo ratio. In fact, a small increase appears to be present in fig 1F, although it is unclear to me if that is significant.

Page 6, third paragraph starting; "Since these results were unexpected..." To me they were expected, based on earlier literature cited above.

Page 8, last paragraph. "Rolling circle DNA synthesis reactions were carried out with T7 replisome wherein one dNTP at a time was replaced with dNTP α S (except dGTP α S). " I wonder, how is the helicase affected by dTTP α S considering that dTTP is the preferred energy source.

Minor point,

Page 11, Figure 7 is discussed after figure 5 and before figure 6. I guess the order of the numbering should be changed.

Suggestions for improving the study (which will be at the author's/editor's discretion)

I think the manuscript would be very much improved if the focus changed from a surprising excessive excision of correct nucleotides, to a starting point where excessive excision of correct nucleotides was earlier demonstrated and here corroborated. The focus should instead be on the mechanism causing excessive excision of correct nucleotides. As such, the title of the study should be changed.

Referee #3:

The paper "Excessive excision of correct nucleotides during DNA synthesis is explained by replication hurdles", by Singh et al. is an exceptionally clear and well-written paper. The experiments are very systematic, logical, and beautifully done. The authors are to be commended for their efforts. The ability of DNA structures to stall primer extension has long been known. Its role in promoting partitioning of correctly paired primer ends to the exonuclease site has also been appreciated in the past. For instance, Reha-Krantz (2014) notes in one of her last papers that "gratuitous proofreading of correctly paired primer ends can be caused by anything that hinders continued primer elongation." But to my knowledge, this story has not been spelled out with the clarity offered here. It is an important contribution. I have just a few comments.

(1) The authors use two well-defined templates: (1) M13 DNA and (2) a circular synthetic template that has a C-rich leading strand and G-rich lagging strand. The asymmetry of this second template make it a useful tool for comparing leading and lagging strand replication by incorporation of α 32P-dGTP or α 32P-dCTP. Thin-layer chromatography allows them to quantify the incorporation of radioactive dGTP into DNA and free dGMP or dCMP, thereby simultaneously comparing incorporation and proofreading on just the leading or lagging strand. I find it curious that the amount of gratuitous proofreading appears to be the same between leading and lagging strand synthesis, given the very different sequence contexts. The authors believe that DNA secondary structure drives proofreading of correct nucleotides on the lagging strand, while polymerase-helicase uncoupling, leading to flap formation, drives proofreading of correct nucleotides on the leading strand. One test of this hypothesis is to assess whether the lagging and leading strand templates have particularly strong pause sites. A second test of the hypothesis is to reverse the leading and lagging strands so that the lagging strand is no long G-rich. In this case, the ratios between polymerization and proofreading may no longer be equivalent on the two strands. If ratios of the two strands remain equivalent, perhaps it is the repetitive nature of the sequence that matters. Slippage of the primer relative to the template at repetitive sequences could create frayed ends that would then be subject to proofreading.

(2) In cells, single-stranded DNA binding proteins play a crucial role in ironing out structures that might otherwise impede DNA replicases. The authors use the minimal replicase from phage T7, including T7 polymerase (gp5), helicase (gp4), and SSB (gp2.5) for most of their experiments. The authors note that gp2.5 is required not only for lagging strand synthesis but also strand-displacement synthesis on the leading strand. It is clear that inclusion of gp2.5 in their M13 assays influences the ratio of polymerization to proofreading. Does increasing the concentration of gp2.5 influence the ratio? In other words, is all ssDNA truly saturated with SSB in their assays?

(3) The energetic costs of this excessive proofreading seem high and it's reasonable to assume that evolution would favor cells that limit this inefficiency. There may be additional factors that limit the formation of these hurdles in cells. In light of that, the authors should discuss how far we can extend these observations to cells. Testing the hypothesis in cells would be challenging. But highly purified complex eukaryotic replication systems have been recently developed by the Diffley lab. Does this represent a way forward?

Minor comments:

(4) Title for figure S3; "exconucleolysis" is misspelled.

(5) In Figure S5D, the production of dGDP is measured over a range of dTTP concentrations. The authors do not describe the purpose of this control experiment, which is presumably to show the extent to which T7 helicase contributes to dGTP hydrolysis under different concentrations of dTTP. The figure legend of figure S5 or the methods would be a suitable place to make that point.

(6) There is no mention of the source of T7 gp2.5 under the subheading "Proteins" in the Methods section.

We thank the reviewers for taking the time for carefully reading our manuscript and providing new insights and constructive feedback to improve the content. We have addressed all the comments and concerns raised by the reviewers by conducting additional experiments and by providing missing details in the text and figures.

Referee #1:

Here, the authors show that, depending on conditions, approx. 2-15% of dNMPs correctly incorporated by T7 DNAP are subsequently excised by "proofreading". They further demonstrate clearly that this activity requires a functional exonuclease site, and that two other replicative polymerases also show this high level of exonucleolysis during normal DNA synthesis. They then proceed to dissect this activity of T7 DNAP under various conditions of limiting dNTP concentrations, etc., to demonstrate conclusively that exonucleolysis is enhanced under conditions where the polymerase may be transiently stalled by replication roadblocks (DNAP inhibited state). Ultimately, they conclusively derive a kinetic model that fits the trends in all of their data.

This is a beautiful, comprehensive, innovative and provocative study, one that challenges current dogma and will have a major impact on our field. **We thank the reviewer for the positive feedback. We also hope that the study will help the field to look beyond the proofreading function of the exonuclease activity of DNAPs.**

My only comments to be considered in revision are minor but should be addressed:

1. Concentrations of some reagents used in particular experiments have been omitted from the Materials and Methods or figure legends. Examples: phi29 DNAP (top of page 21); 30-nucleotide primer (middle of page 21); gene 2.5 protein (middle page 21); trap in legend to Figure 1F; dNTPs in Figure 2A.

Response: Concentrations of phi29 DNAP, protein trap, and dNTPs are now added to the Methods section and Figure legends.

Concentrations of T7 gp2.5 are also added to Table 2. Concentration of Primed M13 ssDNA is mentioned in Table 2 (1:10 ratio of M13 ssDNA and the 30-mer primer was used in annealing reaction).

2. Page 21, typo 5th line from bottom: "formic".

Response: Corrected to "Formic".

3. Figure 1B: Reaction times should be indicated on the figure.

Response: Reaction times are now indicated on top of the TLC image.

4. Figure S5D shows production of dGDP, presumably as a result of use of dGTP by the helicase at limiting dTTP concentrations (should be explained in the legend). Units are not explicitly given, but I assume they relate directly to Figures S5B and C. Figure S5D should be referred to briefly in the text and the origin of the dGTP and units on the y-axis should be clarified in the figure legend. Indeed, the authors should carefully check that all other figures (including those in SI) are referred to in the text.

Response: The reviewer is correct, under limiting dTTP conditions, the helicase will use dGTP and dATP as fuel, hence the observed dGTP hydrolysis. We make a note of this as follows:

“Although the preferred fuel for T7 helicase is dTTP, it can utilize other nucleotides, in particular dATP and dGTP (Pandey & Patel, 2014). Therefore, we lowered the DVTP concentration to 10 μ M and varied the dTTP concentration from 10 μ M to 500 μ M (Figures 6D and Appendix Figure S3A)”.

“We observed the hydrolysis of dGTP to dGTP in our reactions, and as expected, the rate of dGTP hydrolysis decreased with dTTP concentration (Appendix Figure S3D)”.

The units on the Y-axis are now explained in all the Legends. They represent moles of dGDP produced per mole of the replisome complex (which is assumed to be same as the amount of minicircle substrate DNA used in the assay as the limiting reagent in comparison to protein concentrations). Also, we checked to make sure that every figure is referred to in the manuscript.

5. Figures 7 and S6: "Nucleosomes, and other proteins" - nucleosomes are of course irrelevant in bacterial systems, and evidence that they impede replisomes in eukaryotic systems would need to be cited. I would suggest being more general by citing "Protein roadblocks".

Response: Reviewer's concern is well taken. "Nucleosomes and other proteins" is now replaced with "Protein roadblocks" in the final models (Figure 7 and Figure EV5).

6. Figure S3, title typo: exonucleolysis

Response: corrected.

Referee #2:

This is a manuscript with an elegant series of experiments exploring proofreading in the context of the T7 replisome. In the first key experiment the authors monitor the ratio between dNTPs and dNMPs using thin-layer chromatography. To their surprise they found excision of correct nucleotides. Then they follow up this observation by investigating the cause. In brief, they show that replication hurdles, secondary structures in the DNA, slower unwinding of DNA by the helicase, or uncoupled helicase-polymerase, generate frayed primer-ends that are shuttled to the exonuclease site and excised efficiently.

We thank the reviewer for the in-depth review and appreciate the constructive comments.

Major points

The concept of excessive excision of correct nucleotides has already been shown before and has, therefore, to me limited news value. The authors discuss Fersht, et al suggesting that in *E. coli* 7-15% of correct nucleotides are excised. In addition, it was shown by Pavlov, Maki, Maki, and Kunkel (BMC Biology 2004) that 22% of the correctly inserted nucleotides were removed as dTMP by a Family B polymerase in a minimal assay with only the DNA polymerase and a DNA template. The numbers are comparable since thin layer chromatography was used to quantify dTMP and dTTP after the reaction had proceeded for a while. Thus, figure 2 corroborates Pavlov and coworkers finding. Furthermore, the last sentence in the first paragraph on page 7 implying that this is shown for the first time in Family B polymerases needs to be rephrased. Page 7, "these experiments demonstrate that excessive excision of incorporated nucleotides is a general feature of replicative DNA polymerases.

I find the experiments interesting, but disagree with the theme of the paper claiming that excessive excision of correct nucleotides is a novel observation.

What I believe is novel is the approach taken in the investigation of the mechanism causing excessive excision.

Response: We respectfully disagree with the reviewer's comment questioning the novelty of our study.

First, we are reporting excessive excision during ongoing DNA synthesis, which is different from most pol to exo partitioning experiments carried out in the literature, except for those done by Fersht and Reha-Krantz. Second, we carried out studies on circular DNA substrates where excision is not coming from end idling as would be in the experiments pointed out by the reviewer. Third, we have used a fully active reconstituted replisome complex which makes this study physiologically relevant. Fourth, we have carried out a deep mechanistic investigation to understand this new "proofreading activity". We argue that all combined makes this study significant and novel.

It is well known that when the DNAP reaches the end of a linear replication template, it remains bound to the primer-end and idles, which produces excessive amounts of dNMPs. The DNA substrates used in Pavlov et al, 2004 were linear polydA/dT, and these are steady state experiments, the main source of excessive excision would be from DNAP idling at the primer-end after completion of the reaction. What we report here are experiments with circular DNA substrates that support rolling circle DNA synthesis and hence do not have the end-problem. We concluded that under these conditions, excessive excision of nascent DNA arises from frequent primer-end shuttling between pol and exo site due to translocation hurdles such as DNA

secondary structures, polymerase-helicase uncoupling, etc. We suggested that primer-end partitioning to the exo-site could be beneficial in preventing mutagenic extensions, but the cost for this type of proofreading is high.

Specific points:

Page 3, "Based on the excision rates of correctly base-paired 3'-end nucleotide, this partitioning model predicts that only 0.1% of the correctly incorporated nucleotide will be excised during DNA synthesis". As, I perceive the manuscript, this is why the authors are surprised that the T7 DNA polymerase and other DNA polymerases perform excessive excision of correct nucleotides. Could it be that the model was incorrect and if so can you today clarify why that was the case?

Response: The reviewer correctly understands the motivation of our study. When we observed excessive excision on circular substrates, it raised our curiosity. What intrigued us is the seeming discrepancy between our results and those predicted by the exonuclease partitioning model which was proposed by my (Smita Patel) own work on T7 DNAP. According to the partitioning model, the correctly base-paired primer-end rarely partitions to the exonuclease site during ongoing DNA synthesis, but the present experiments were suggesting otherwise.

The conclusion from the present study is that the partitioning model is correct, in that under ideal DNA synthesis conditions, where the DNAP does not encounter any translocation hurdles, exo visitations would be rare. However, when DNAP is synthesizing long stretches of DNA, it inevitably encounters many translocation hurdles during leading and lagging strand synthesis that increases the exo visitations, some of which are summarized in our model in Figure 7.

Page 5, third paragraph, In all experiments are very high concentrations of deoxyribonucleotides used. In fact, they are much higher than the determined in vivo concentrations. Could the outcome of experiments exploring the effect of dNTP concentration been different if dNTP concentrations in the range 1-11 μM was used?

Response: The dNTP concentrations in the assays with primed M13 ssDNA (Figure 5) were indeed in the range suggested by the reviewer from 1 μM to 30 μM . The Pol/Exo ratio was low (~5) at low dNTPs and increased (~35) at higher dNTPs concentrations.

Furthermore, would adding ribonucleotides at mM concentration also affect the balance between the exonuclease and polymerase activities?

This is an interesting question, so we performed leading strand synthesis reactions on the minicircle substrate with 1 mM ATP. We observed decreased Pol-activity, increased Exo-activity, and a lower Pol/Exo to around 5 in comparison to reactions without ATP. These are intriguing results that we hope to pursue in the future, hence not included in the present study.

Figure for referees not shown.

The dNTP concentration is a very important parameter in relation to the presented model on page 17; "This spontaneous partitioning model (Figure 7, blue-box) predicted that the Exo-activity should decrease with increasing dNTP concentrations (Figure 5E and 6B, gray lines). However, our measurements show that the Exo-activity remains constant and does not reduce with increasing dNTP levels (Figure 5E and 6B). We, therefore, concluded that excessive excision activity is not due to spontaneous shuttling of the primer-end into the Exo-site. In other words, it is not the "cost of proofreading" as suggested previously."

Perhaps the exonuclease activity decrease with increasing dNTP concentrations if the starting concentration is much lower. That was shown to be the case in figure 8 by Ganai et al, NAR 2015. How would measurements at much lower dNTP concentrations influence the final model that depends on the generation of frayed 3'-termini?

Response: Note that the experimental setup used by Ganai et al does not allow measurement of excision activity during ongoing DNA synthesis. They measured excision of the original primer-end whereas we are measuring excision of the incorporated dNMPs. Thus, our Pol to Exo ratio reflects partitioning in the extension phase. Spontaneous partitioning model predicts a decrease in exo-activity with increasing dNTP concentrations, which we did not observe in our experiments. As explained above, we did not see a decrease in exo-activity even in the lower concentration range of dNTPs.

Page 5, last sentence on the page, "If the increased excisions were due to DNAP dissociating from the DNA substrate and rebinding of the primer-end preferentially in the Exo-site, the Pol/Exo ratio would decrease with the protein trap. The Pol/Exo ratio remained unchanged after the addition of the trap (Figure 1F, Figure S1C)." Would not the binding of DNAP to a primer with a 3'-ddNMP in the presence of the high dNTP concentration freeze the DNA in the polymerase site. Thus, I would expect a decrease in exonuclease activity and an increased Pol/Exo ratio. In fact, a small increase appears to be present in fig 1F, although it is unclear to me if that is significant.

Response: We agree with the reviewer that the trap with the 3'ddNMP end is an effective trap that would stably remove free and newly dissociated DNAPs without itself being a source of excision products. The slight increase in Pol/Exo ratio in presence of protein trap is not significant.

For clarity, we have changed the wording:

"To determine if the excision was occurring during processive DNA synthesis and was not due to protein dissociation and rebinding

events, we added a protein trap (35/68 mer primer-template with 3'dideoxy primer-end) 30 s after starting the rolling circle reaction with the T7 replisome. The trap with the 3'ddNMP end is an effective trap that would stably remove free and newly dissociated DNAPs without itself being a source of excision products. The Pol/Exo ratio remained unchanged after addition of the trap (Figure 1F, Figure EV1A). The unchanged Pol/Exo ratio indicates that excessive excision is occurring during processive DNA synthesis, hence the primer-end transfer from the Pol-site to the Exo-site is intramolecular."

Page 6, third paragraph starting; "Since these results were unexpected..."
To me they were expected, based on earlier literature cited above.

Response: Please see our explanation above.

Page 8, last paragraph. "Rolling circle DNA synthesis reactions were carried out with T7 replisome wherein one dNTP at a time was replaced with dNTP α S (except dGTP α S). " I wonder, how is the helicase affected by dTTP α S considering that dTTP is the preferred energy source.

Good point, we conducted new experiments. Interestingly, our stopped-flow helicase unwinding assays show that the rates of fork unwinding are the same with dTTP and dTTP α S. We believe these results will be of interest and hence added the experiments in Supplementary Figures (Figures EV3Bi and EV3Bii) with the following lines in the manuscript:

"T7 helicase uses both dTTP and dTTP α S equally as fuel for DNA unwinding (Figures EV3Bi and EV3Bii). This allowed us to perform experiments with dTTP α S."

The details of the method used have been incorporated in the Methods section.

Minor point,

Page 11, Figure 7 is discussed after figure 5 and before figure 6. I guess the order of the numbering should be changed.

Response: We considered this change, but it hampered with the flow in the description in the results section, hence we have decided to keep the figure order unchanged.

Suggestions for improving the study (which will be at the author's/editor's discretion)

I think the manuscript would be very much improved if the focus changed from a surprising excessive excision of correct nucleotides, to a starting point where excessive excision of correct nucleotides was earlier demonstrated and here corroborated. The focus should instead be on the mechanism causing excessive excision of correct nucleotides. As such, the title of the study should be changed.

Response: The current title says that excessive excision is due to replication hurdles and hence is in line with the reviewer's suggestion.

Referee #3:

The paper "Excessive excision of correct nucleotides during DNA synthesis is explained by replication hurdles", by Singh et al. is an exceptionally clear and well-written paper. The experiments are very systematic, logical, and beautifully done. The authors are to be commended for their efforts. The ability of DNA structures to stall primer extension has long been known. It's role in promoting partitioning of correctly paired primer ends to the exonuclease site has also been appreciated in the past. For instance, Reha-Krantz (2014) notes in one of her last papers that "gratuitous proofreading of correctly paired primer ends can be caused by anything that hinders continued primer elongation." But to my knowledge, this story has not been spelled out with the clarity offered here. It is an important contribution. I have just a few comments.

(1) The authors use two well-defined templates: (1) M13 DNA and (2) a circular synthetic template that has a C-rich leading strand and G-rich lagging strand. The asymmetry of this second template make it a useful tool for comparing leading and lagging strand replication by incorporation of $\alpha^{32}\text{P}$ -dGTP or $\alpha^{32}\text{P}$ -dCTP. Thin-layer chromatography allows them to quantify the incorporation of radioactive dGTP or dCTP into DNA and free dGMP or dCMP, thereby simultaneously comparing incorporation and proofreading on just the leading or lagging strand. I find it curious that the amount of gratuitous proofreading appears to be the same between leading and lagging strand synthesis, given the very different sequence contexts. The authors believe that DNA secondary structure drives proofreading of correct nucleotides on the lagging strand, while polymerase-helicase uncoupling, leading to flap formation, drives proofreading of correct nucleotides on the leading strand. One test of this hypothesis is to assess whether the lagging and leading strand templates have particularly strong pause sites. A second test of the hypothesis is to reverse the leading and lagging strands so that the lagging strand is no long G-rich. In this case, the ratios between polymerization and proofreading may no longer be equivalent on the two strands. If ratios of the two strands remain equivalent, perhaps it is the repetitive nature of the sequence that matters. Slippage of the primer relative to the template at repetitive sequences could create frayed ends that would then be subject to proofreading.

Response: We thank the reviewer for the positive comments and for carefully reading the manuscript.

We were also intrigued by the similar Pol/Exo values during leading and lagging strand synthesis and believe that it is a coincidence. We believe that changing the minicircle DNA sequence could result in a different Pol/Exo ratio for the two strands. We plan to check this out after we make new minicircle templates.

Thank you for the suggestion for looking into pause sites. We did find the presence of an unintentional 6 bp secondary structure in the lagging strand, which could be a hurdle to the translocation of the

lagging strand DNAP. We have added the DNA secondary structure to Figure EV1B (Figure EV1Biii).

As we were thinking about your comment, we realized that another source of excision activity during lagging strand synthesis is idling in the period between the completion of one Okazaki synthesis and the start of another.

We have included these points in the results:

“The Pol/Exo ratio during lagging strand synthesis was 16 ± 3 (Figure 1H), which is similar to the ratio during leading strand synthesis. We believe that the similarity in the ratios is coincidental and they could be different with a minicircle DNA of a different sequence”.

And in the discussion:

“We observed that the Pol/Exo ratio during lagging strand synthesis reactions on minicircle DNA substrate is around 16, which is much lower than the Pol/Exo ratio of ~ 35 from primer extension reactions on primed M13 ssDNA. Similar to the conclusions drawn from the M13 ssDNA experiments, the source of excessive excision during lagging strand synthesis could be the presence of DNA secondary structures in the lagging strand template (Figure EV1Biii). Furthermore, when lagging strand DNAP completes the synthesis of an Okazaki fragment, it encounters the 5'-end of the previous Okazaki fragment and remains bound to the primer-end. Under these conditions, the DNAP can idle between Pol- and Exo-sites until it recycles to a newly synthesized primer (Garg et al., 2004). We speculate that DNAP idling in the time window between two Okazaki fragment synthesis contributes to excessive excision during lagging strand synthesis. Additionally, as a component of the replisome, the lagging strand DNAP interacts with the N-terminal domain of T7 helicase-primase (Gao, Cui et al., 2019). Such interactions, especially those involving the exonuclease domain of DNAP could alter the dynamics of active site switching”.

(2) In cells, single-stranded DNA binding proteins play a crucial role in ironing out structures that might otherwise impede DNA replicases. The authors use the minimal replicase from phage T7, including T7 polymerase (gp5), helicase (gp4), and SSB (gp2.5) for most of their experiments. The authors note that gp2.5 is required not only for lagging strand synthesis but also strand-displacement synthesis on the leading strand. It is clear that inclusion of gp2.5 in their M13 assays influences the ratio of polymerization to proofreading. Does increasing the concentration of gp2.5 influence the ratio? In other words, is all ssDNA truly saturated with SSB in their assays?

Response: Based on the reviewer’s comments, we performed experiments at a higher concentration of T7 gp2.5. Doubling T7 gp2.5 concentration did not change the pol- and exo-activities. This confirms that the M13 ssDNA is fully saturated with the T7 gp2.5 protein. The results are included in Figure 5 and following sentence is added to the results section:

“Indeed, the Pol/Exo ratio in the absence of gp2.5 was 2.1 ± 0.1 in comparison to 33 ± 3 in the presence of gp2.5 (Figure 5H and Appendix Figure S2). These results indicate that secondary structures

in template DNA are responsible for the majority of correct nucleotide excision during DNA synthesis on M13 ssDNA. Based on T7 gp2.5 footprint of ~7 nucleotides on ssDNA (Hernandez & Richardson, 2019), the gp2.5 concentrations were sufficient to saturate the M13 ssDNA (Table 2). Doubling the concentration of gp2.5 protein did not change the Pol- and Exo-activities (Figure 5H and Appendix Figure S2)”

(3) The energetic costs of this excessive proofreading seem high and it's reasonable to assume that evolution would favor cells that limit this inefficiency. There may be additional factors that limit the formation of these hurdles in cells. In light of that, the authors should discuss how far we can extend these observations to cells. Testing the hypothesis in cells would be challenging. But highly purified complex eukaryotic replication systems have been recently developed by the Diffley lab. Does this represent a way forward?

Response: Certainly, it is important to test this result under cellular conditions. But we agree with the reviewer that it could be challenging. Carrying out similar studies with the *in vitro* reconstituted eukaryotic replication complex is possible. We take note of these ideas and point it out in the Discussion section:

“Interestingly, the energetic cost of protecting the primer-end from mutagenic extension is much greater than that of proofreading misincorporations. It is possible that there are additional cellular factors that resolve these replication hurdles and reduce the energetic cost of primer-end protection. In contrast to T7 replisome, eukaryotic replisomes are complex and constitute of many protein factors (O'Donnell, Langston et al., 2013, Pellegrini & Costa, 2016). There have been successes in reconstituting the eukaryotic replisome *in vitro* (Taylor & Yeeles, 2018, Yeeles, Janska et al., 2017). Hence, the extent of excision during DNA synthesis by the eukaryotic replisome and the effect of the individual components on the proofreading cost can be studied in the future”.

Minor comments:

(4) Title for figure S3; "exconucleolysis" is misspelled.

Response: Spelling corrected.

(5) In Figure S5D, the production of dGDP is measured over a range of dTTP concentrations. The authors do not describe the purpose of this control experiment, which is presumably to show the extent to which T7 helicase contributes to dGTP hydrolysis under different concentrations of dTTP. The figure legend of figure S5 or the methods would be a suitable place to make that point.

Response: We thank the reviewer to pointing this out. We have incorporated following sentences in the manuscript:

“Although the preferred fuel for T7 helicase is dTTP, it can utilize other nucleotides, in particular dATP and dGTP (Pandey & Patel, 2014). Therefore, we lowered the DVTP concentration to 10 μ M and varied the dTTP concentration from 10 μ M to 500 μ M (Figures 6D and Appendix Figure S3A)”.

“We observed the hydrolysis of dGTP to dGDP in our reactions, and as expected, the rate of dGTP hydrolysis decreased with dTTP concentration (Appendix Figure S3D)”.

(6) There is no mention of the source of T7 gp2.5 under the subheading "Proteins" in the Methods section.

Response: A reference has now been added to cite the purification method used to prepare T7 gp2.5.

2nd Editorial Decision

29th November 2019

Thank you for submitting your revised manuscript for our consideration. Two of the original referees have now assessed it once more, and given that they were generally satisfied with your responses and revisions, we shall be happy to accept the study for publication in The EMBO Journal, pending final minor revision of a few remaining textual points mentioned by the referees. In particular, please make sure to incorporate the discussions currently only present in the response letter, as requested by referee 3, into the final text. For our easier assessment of the final changes, I would appreciate if you used the "Track Changes" option when editing these parts.

REFeree REPORTS

Referee #1:

In my opinion, the authors have addressed all of the scientific and methodological comments of the Reviewers, and this paper should be accepted for publication asap.

It is an influential contribution that will inform key papers in the field well into the future.

I have editorial suggestions to incorporate at the earliest opportunity:

Page 12, para. 2, line 4: "from the dNTP bound post-translocated state"

Page 15, first line: "hydrolysis of dGTP to dGDP in our reactions"

Referee #3:

The manuscript by Singh et al. presents a provocative set of mechanistic experiments with purified T7 replication components (and validated with other polymerases), which reveals a large proportion of correctly inserted nucleotides are removed by the proofreading exonuclease during both leading and lagging strand DNA synthesis. These events do appear to be spontaneous partitioning of the primer end to the proofreading domain. Rather they are instigated by impediments to polymerization such as secondary structures that create transient frayed ends. This work complements recent findings that indicate that a key proofreading-defective cancer allele (POLE-P286R) likely increases mutagenesis by causing defects in binding of the primer end to the exonuclease domain. This excellent work should have an impact on the field and is of great interest.

The authors have been very responsive to the comments of the reviewers and addressed many of our concerns with additional experiments, changes to the text, or with robust counter arguments. They should be congratulated on a beautiful story. Hopefully it will provoke creative ways of exploring

whether this energetically costly mechanism is at work within cells.

I have only one additional comment:

Reviewer 2 questioned the novelty of this finding based on experiments that came to a similar conclusion (perhaps for the wrong reason). The authors counter on page 3 in their response with a reasonable argument. But they don't modify the text in anyway. Other readers may have a similar question. For this reason, I suggest that they include a statement at the end of Paragraph 1 of the Results at the top of page 5 that states how using a circular template and polymerases that do strand displacement synthesis eliminates the end-idling problem that would confound their observations if made with linear templates. And they should give a reference for the paper that demonstrates end-idling is a problem. This one sentence would educate the reader on something they may not know and further strengthen the logic of using this approach.

2nd Revision - authors' response

23rd December 2019

Referee #1:

In my opinion, the authors have addressed all of the scientific and methodological comments of the Reviewers, and this paper should be accepted for publication asap.

It is an influential contribution that will inform key papers in the field well into the future.

I have editorial suggestions to incorporate at the earliest opportunity:

Page 12, para. 2, line 4: "from the dNTP bound post-translocated state"

Page 15, first line: "hydrolysis of dGTP to dGDP in our reactions"

Response: We thank the reviewer for strongly positive review of our work. We also thank the reviewer for pointing out the mistakes in the manuscript. Both the errors are now rectified in the revised manuscript.

Referee #3:

The manuscript by Singh et al. presents a provocative set of mechanistic experiments with purified T7 replication components (and validated with other polymerases), which reveals a large proportion of correctly inserted nucleotides are removed by the proofreading exonuclease during both leading and lagging strand DNA synthesis. These events do appear to be spontaneous partitioning of the primer end to the proofreading domain. Rather they are instigated by impediments to polymerization such as secondary structures that create transient frayed ends. This work complements recent findings that indicate that a key proofreading-defective cancer allele (POLE-P286R) likely increases mutagenesis by causing defects in binding of the primer end to the exonuclease domain. This excellent work should have an impact on the field and is of great interest. The authors have been very responsive to the comments of the reviewers and addressed many of our concerns with additional experiments, changes

to the text, or with robust counter arguments. They should be congratulated on a beautiful story. Hopefully it will provoke creative ways of exploring whether this energetically costly mechanism is at work within cells.

I have only one additional comment:

Reviewer 2 questioned the novelty of this finding based on experiments that came to a similar conclusion (perhaps for the wrong reason). The authors counter on page 3 in their response with a reasonable argument. But they don't modify the text in anyway. Other readers may have a similar question. For this reason, I suggest that they include a statement at the end of Paragraph 1 of the Results at the top of page 5 that states how using a circular template and polymerases that do strand displacement synthesis eliminates the end-idling problem that would confound their observations if made with linear templates. And they should give a reference for the paper that demonstrates end-idling is a problem. This one sentence would educate the reader on something they may not know and further strengthen the logic of using this approach.

Response: We would like to thank the reviewer for the positive and in-depth review of the manuscript. As suggested by the reviewer, we have now included the following sentences in the Results:

“To simultaneously measure the polymerase (Pol) and exonuclease (Exo) activities, we chose a 70 bp minicircle DNA annealed to a 110-mer primer that supports efficient rolling circle leading and lagging strand DNA synthesis (Lee, Chastain et al., 1998, Pandey, Syed et al., 2009). Previous studies that measured Pol- and Exo-activities used linear primer-templates where the polymerase would idle at the 3'-end of the primer and produce the dNMP excision products (Mizrahi, Benkovic et al., 1986, Pavlov, Maki et al., 2004). Use of circular minicircle template in the present study eliminates the end-idling problem and provides an accurate measurement of the Exo-activity during active DNA synthesis.”

Accepted

7th January 2020

Thank you for submitting your final revised manuscript for our consideration. I am pleased to inform you that we have now accepted it for publication in The EMBO Journal.

Corresponding Author Name: Smita S Patel

Journal Submitted to: The EMBO Journal

Manuscript Number: EMBOJ-2019-103367R